**Brief Communication**

# Fast and robust metagenomic sequence comparison through sparse chaining with skani

**Jim Shaw** 🔘 [1] ✉ **& Yun William Yu** 🔘 [1,2,3] ✉

Sequence comparison tools for metagenome-assembled genomes (MAGs) struggle with high-volume or low-quality data. We present skani (https://github.com/bluenote-1577/skani), a method for determining average nucleotide identity (ANI) via sparse approximate alignments. skani outperforms FastANI in accuracy and speed (>20× faster) for fragmented, incomplete MAGs. skani can query genomes against >65,000 prokaryotic genomes in seconds and 6 GB memory. skani unlocks higher-resolution insights for extensive, noisy metagenomic datasets.

Consider the fundamental problem of computing sequence-to-sequence similarity between metagenome-assembled genomes (MAGs). Modern studies generate hundreds of thousands of MAGs[1,2], and searching these MAGs against a database or computing all pairwise similarities takes billions of comparisons; this is infeasible with traditional alignment-based methods. Thus, large-scale sequence comparison for metagenomic data is dominated by sketching methods. Sketching methods summarize datasets into small collections of k-mers; these sketches can be efficiently compared against one another and return an average nucleotide identity (ANI) estimate.

Unfortunately, sketching methods such as Mash[3] or sourmash[4] may underestimate ANI when genome incompleteness is present[5]. The decrease has nothing to do with the *genetic distance* between the genomes; it is simply an *artefact* of the assembly being incomplete. Even 'medium-quality' MAGs typically only require >50% completeness[1], so this is an issue in practice. On the other hand, alignment-based methods are able to estimate ANI from only the orthologous regions, so incompleteness is not an issue. Additionally, the fraction of the genomes aligned to one another (the aligned fraction) is a useful statistic that pure sketching methods do not estimate. There is thus a need for algorithms that are fast, like sketching methods, yet robust to noise due to assembly artefacts, like alignment methods.

We developed skani, a fast, robust tool for calculating aligned fraction and ANIs in the >82% range. skani's ANI method is robust against incomplete and fragmented MAGs, yet it is multiple orders of magnitude faster than alignment-based methods and over an order of magnitude faster than even the state-of-the-art FastANI[6]. skani uses a very sparse k-mer chaining[7–9] procedure to quickly find orthologous regions between two genomes. This allows for sequence identity estimation using k-mers on only the shared regions between two genomes (Fig. 1a), avoiding the pitfalls of alignment-ignorant sketching methods. Like BLAST-based ANI methods, skani breaks genomes into nonoverlapping fragments, estimates the ANI for each fragment and then averages the ANI to output an ANI estimate. We then use a trained regression model to debias our ANI estimates (Methods).

We first verified that existing ANI methods are indeed sensitive to incompleteness and fragmentation in Extended Data Fig. 1. In a synthetic test, fragmented, incomplete yet identical genomes had ANI estimates that were systematically lower than 100% for all methods but ANIm[10,11], a slow but accurate method. We subsequently chose ANIm as a baseline when comparing MAGs. Mash was the most affected, with up to a difference of 4% ANI at 50% completeness, which can cause two very similar genomes of up to 99% ANI to be classified as different species when subject to the standard 95% ANI species threshold[6]. FastANI was sensitive to fragmentation (low N50), which is why a minimum N50 of 10,000 is used in the original study[6], but that N50 requirement is not met in many real experiments[1,12]. We additionally show simulations with mutations and chimeric genomes in Supplementary Figs. 1–3.

Next, we showed on real MAGs that only skani and ANIm are robust to MAG quality for high-resolution ANI calculations. In Fig. 1b, we compared subspecies level MAGs generated by Pasolli et al[1], a large collection of medium-quality and high-quality short-read assembled MAGs.

[1]Department of Mathematics, University of Toronto, Toronto, Ontario, Canada. [2]Computer and Mathematical Sciences, University of Toronto at Scarborough, Toronto, Ontario, Canada. [3]Computational Biology Department, Carnegie Mellon University, Pittsburgh, PA, USA. ✉e-mail: jshaw@math.toronto.edu; ywyu@cmu.edu

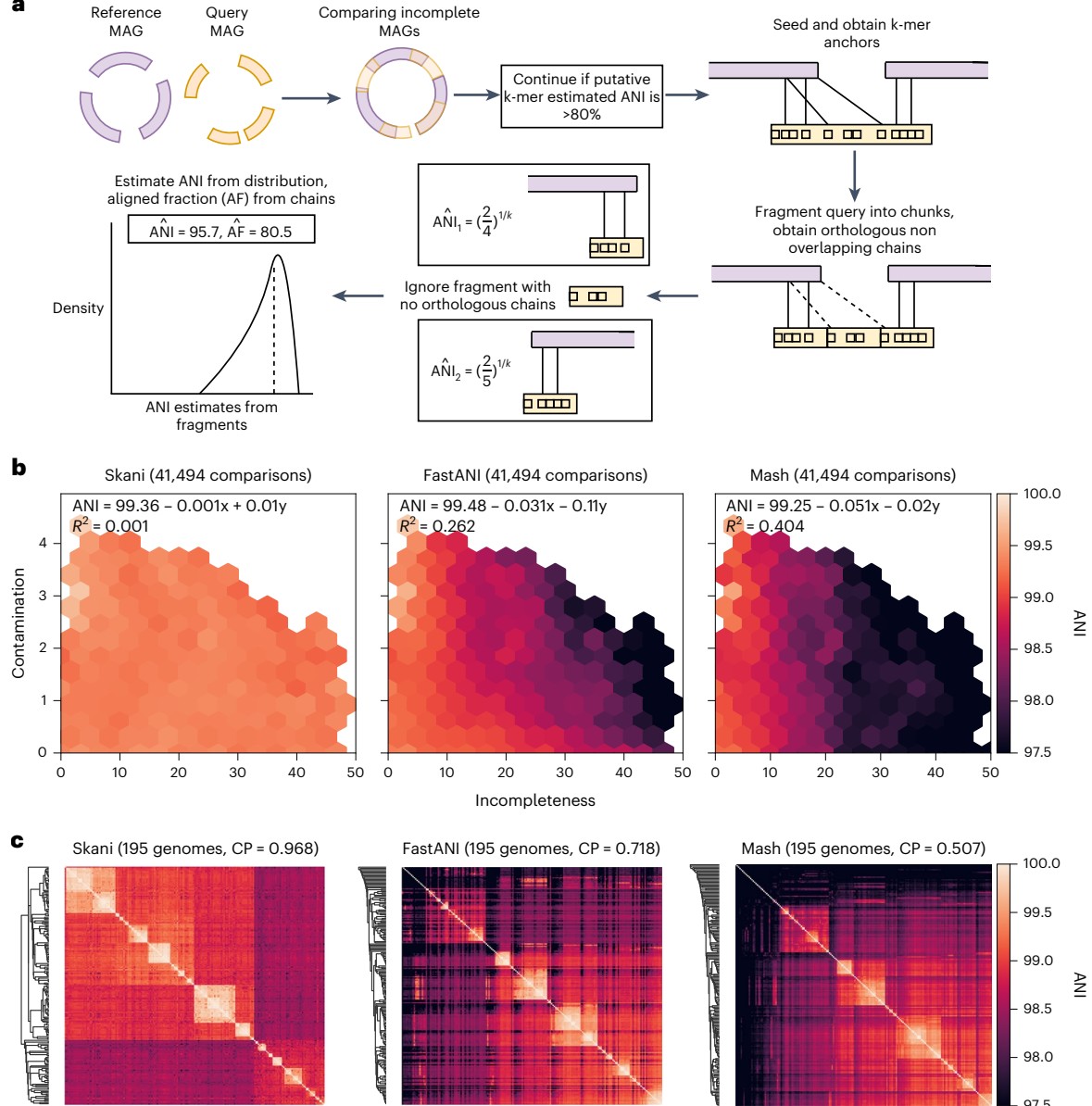

**Fig. 1 | a, Algorithm overview of skani. b,** ANI sensitivity to contamination and incompleteness. We took all pairs of MAGs with >99% ANI according to ANIm from species-level bins generated by Pasolli et al.[1] with >25 and <50 genomes, leading to a diverse set of 41,494 pairs of genomes. We re-evaluated the ANI of each method and performed ordinary least-squares regression with incompleteness and contamination (averaged between the pair and obtained by CheckM[22]) as covariates. Estimated parameters and $R^2$ values are shown; only hexagons with >20 data points are visible (see Supplementary Fig. 4 for density information). **c,** Average-linkage cluster heatmap for each method on bin number 2328 from Pasolli et al.[1] (classified as *Alistipes ihumii*) with 195 genomes. Cophenetic correlation (CP) of each method's dendrogram (with ANIm's distance matrix as a ground truth) is shown. skani's high cophenetic correlation indicates that its dendrogram is concordant with ANIm's dendrogram, which we show in Extended Data Fig. 2. The Robinson-Foulds distances[23] for skani, FastANI and Mash's dendrograms against ANIm's average-linkage tree are 0.489, 0.713 and 0.823, respectively.

As incompleteness and contamination increase, Mash and FastANI trend toward lower ANI, but this is not seen for ANIm (Extended Data Fig. 2) or skani. Extended Data Fig. 3 confirms our simulation results and shows that incompleteness and fragmentation are to blame for the bias.

Because ANI underestimations due to MAG quality are systematically biased, such ANI estimates can strongly impact downstream applications. We show in Fig. 1c that the cluster heatmaps obtained by average-linkage clustering for a species-level bin differ greatly between ANI methods. skani's heatmap qualitatively resembles ANIm's heatmap (Extended Data Fig. 2) more closely than the other methods, yet it is >500 times faster than ANIm and >50 times faster than FastANI for computing the distance matrix (Supplementary Fig. 5).

To quantify the concordance of the clustering against ANIm, we used cophenetic correlation[13] (Supplementary Note). In Fig. 1c and Extended Data Fig. 4, we see that skani has a better cophenetic correlation with respect to ANIm than all other compared methods. Notably, the ordering of the methods with respect to cophenetic correlation in Extended Data Fig. 4 is skani > sourmash max-contain[14] > FastANI > mash; this exactly agrees with the ordering using $R^2$ values from the contamination-incompleteness plots in Fig. 1b and Extended Data Fig. 2, implying that MAG assembly artefacts are indeed to blame for the clustering discordance.

We explore skani on three additional datasets: ocean eukaryotic MAGs[15,16], ocean archaea MAGs[12] and soil prokaryotic MAGs[17], which

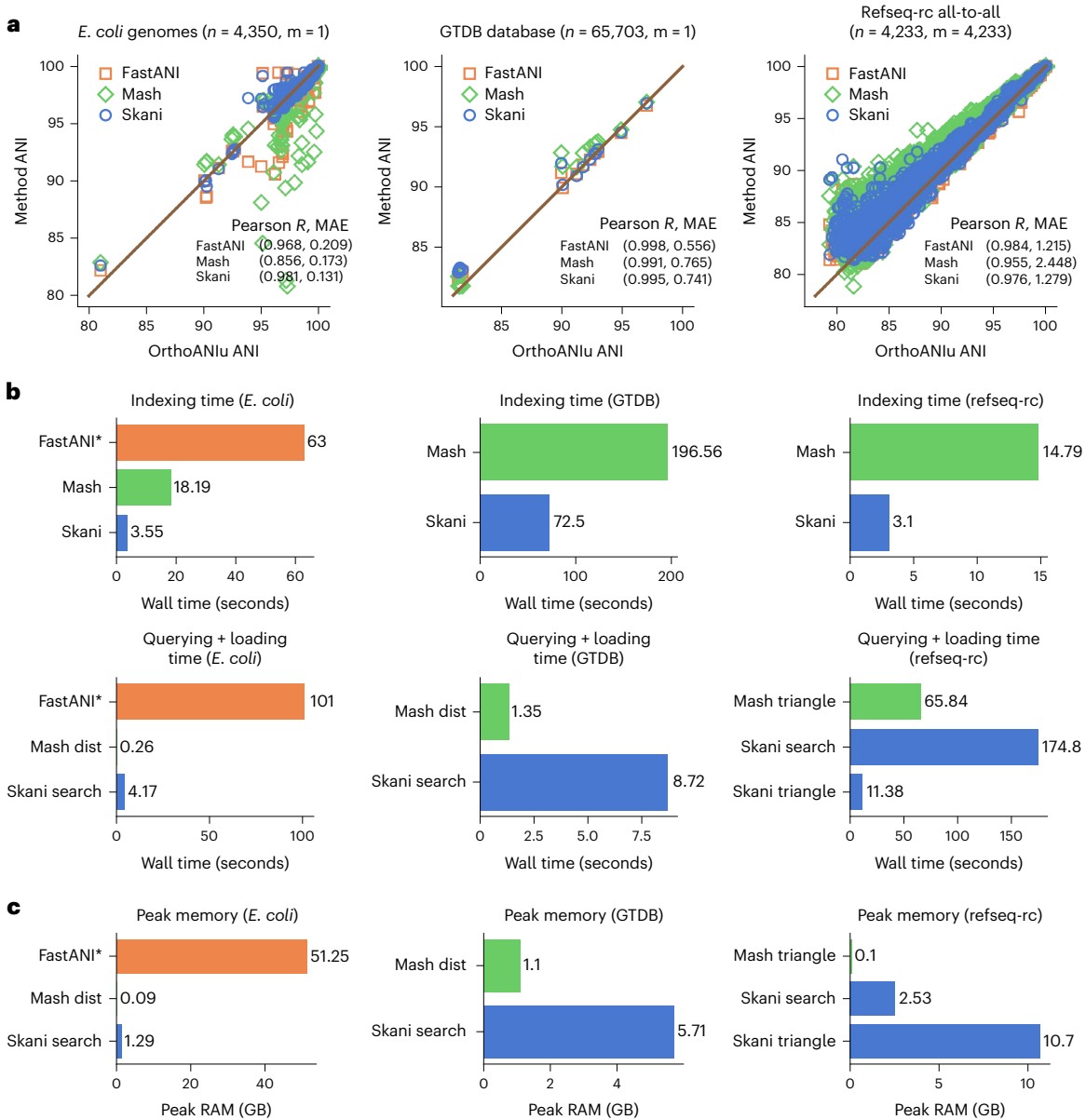

**Fig. 2 | a, ANI benchmarking with *n* reference genomes and *m* query genomes.** From left to right: (1) querying a single E. coli K12 genome against a collection of E. coli genomes, (2) querying a single *E. coli* K12 genome against the GTDB R207 database and (3) all-to-all comparisons on the refseq-rc (representative and complete bacterial genomes) database. OrthoANIu[20] is used as a baseline. We only analyzed data points for which all methods had a predicted value. Pearson R value and mean absolute error (MAE) are shown for each dataset. Dataset descriptions can be found in Supplementary Table 2. **b,** Indexing and querying wall time for each dataset (50 threads). See Supplementary Fig. 8 for CPU times instead. Subcommands used are shown for each method when applicable. FastANI indexing and query times were estimated from the output. skani search and skani triangle are different subcommands that give the same results, but skani search only loads genomes into RAM as needed and discards after usage. FastANI times are not shown for the GTDB and refseq-rc datasets for fairness to FastANI due to FastANI's ability to output a slightly larger range of ANI values (approximately >75% for FastANI versus >82% for skani). **c.** Peak memory usage for each method and subcommand. Sketching took negligible memory for skani and Mash.

in addition to the Pasolli et al. dataset gives four datasets in total. Extended Data Fig. 5 shows that skani's results generalize to a diverse set of genomes, including eukaryotic MAGs with a median size of 17.6 Mbp (and 32 MAGs of size >100 Mbp). When comparing each method's deviation from ANIm and considering the 1 to 99 percentile deviations, skani has the smallest 1 to 99 percentile interval lengths, indicating robustness. Extended Data Figs. 6 and 7 show that skani has a better linear aligned fraction correlation with ANIm than FastANI. The aligned fraction accuracy can be improved further by controlling subsampling rate of the k-mers (parameter *c* in Methods).

An important task is classifying MAGs (or isolate genomes) by searching against a database of reference genomes. Such databases represent a diverse collection of genomes where only a fraction of the genomes are similar to the query. Therefore, sensitively searching against each reference is unnecessary. To enable efficient database search, we augmented skani with a quick sketching-based ANI filter against distant genomes before performing a more accurate ANI computation.

Figure 2 shows that skani can query an *Escherichia coli* genome against the GTDB R207 database[18] (>65,000 genomes) in comparable speed and memory to Mash. skani is much faster than FastANI for querying (>20 times on the *E. coli* dataset) and is >2.5 times faster than Mash for indexing. Furthermore, skani can do all-to-all comparisons on a set of 4,233 bacterial genomes as quickly as Mash due to the fast

filtering of unrelated genomes. FastANI takes much longer (Supplementary Fig. 9), but FastANI does more comparisons due to a slightly larger valid ANI range.

In Fig. 2a and Supplementary Fig. 7, we benchmarked skani against OrthoANIu[19,20] (which we shorten to ANIu), a faster but almost-identical analogue of the BLAST-based ANIb, as a baseline. We did not use ANIm, because ANIm overestimates ANI for pairs of genomes with <90% ANI[20,21]. skani outputs an ANI estimate only if one of the genomes has predicted AF ≥15% by default, which ends up giving reasonable ANIs down to the 82% range on the three datasets shown. skani's accuracy is usually better than Mash but slightly worse than FastANI for reference-quality genomes, skani is better on the *E. coli* dataset, which includes many fragmented, possibly incomplete genomes that give rise to many Mash and FastANI outliers. The results for FastANI improved (Pearson $R$ from 0.974 to 0.994) on the E. coli dataset if we removed genomes with N50 <10,000, giving the exact same Pearson $R$ value as the originally reported FastANI results[6]. Thus, skani gives only slightly less accurate values than FastANI on reference-quality genomes with the assurance of robustness for low-quality assemblies.

We have shown that skani improves on the state-of-the-art for metagenomic sequence comparison. skani's key operating regimes are for medium-to-high ANI (>82%), comparisons against diverse sets of genomes (such as databases), and fast all-to-all comparisons for up to tens of thousands of highly similar genomes. skani is limited in extreme regimes, such as low ANI or comparing hundreds of thousands of similar genomes (for example all-to-all calculation for the >200,000 *E. coli* genomes currently available in RefSeq). Future work includes exploring parameter choices and methods for accessing more extreme operating regimes, for example, linear-time clustering heuristics or more sensitive amino-acid computations.

In conclusion, skani is almost as fast as sketching-based methods for ANI database search, yet it gives a more robust signal when comparing noisy MAGs. Given the overwhelming amount of data generated by modern metagenomic studies, we believe skani's ability to analyze an order of magnitude more data while simultaneously giving a stronger signal will allow examination of vast metagenomic sequences at a higher resolution, unlocking new types of analysis not possible before.

## Online content

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

## Methods

### Sequence identity estimation

Formally, let $G$ be a string of nucleotides and $G'$ be a mutated version of $G$ where every letter is independently changed to a different letter with probability $\theta$. We will define the true ANI to be equal to $1 - \theta$ under our model. Under the usual assumption of no repetitive k-mers[24], it is easy to estimate $\theta$ from k-mer matching statistics[3,24,25] between $G$ and $G'$. We proved in a previous work that for random, mutating strings, the expected number of k-mer matches arising spuriously from repetitiveness for a string of length $n$ is $\sim \frac{n^2}{4^k}$ (Theorem 4 in Shaw and Yu[26]), so the usual assumption of no repetitive k-mers is not a bad one in practice for simpler, non-eukaryotic genomes and large enough $k$.

Sketching methods such as Mash[3] or sourmash[4] use the above framework to calculate ANI through the aforementioned k-mer statistics, which are built around estimating the normalized overlap between the k-mers in two genomes. More precisely, these *indices* take the form $\frac{|A \cap B|}{f(A,B)}$ where $A$ and $B$ are the sets of k-mers in two genomes, and $f(A, B)$ controls the normalization. The Jaccard index used in Mash[3] corresponds to $f(A, B) = |A \cup B|$, and the containment index is $f(A, B) = |A|$ (ref. 14). The main index we consider is the max-containment index, which corresponds to $f(A, B) = \min(|A|, |B|)$; minimizing the denominator maximizes the containment. These indices can approximate ANI through simple formulas[25,27].

However, when dealing with MAGs, we do not have $G$ and $G'$ but instead fragmented, contaminated and incomplete versions of $G$ and $G'$. The models used in these sketching methods give biased estimates for ANI, resulting in underestimated ANIs, because missing k-mer matches may be due to mutations, incompleteness, or contamination instead of only due to mutations. Thus, all of the above k-mer statistics suffer from the following problem: as two MAGs become more incomplete, the overlap $|A \cap B|$ may decrease more than $f(A, B)$ because a homologous k-mer is simply not present in the assembly, therefore making the index smaller.

This issue is also present in the context of k-mer based alignment-free genome comparison using reads[28–30], where a genome could potentially not be fully covered due to the random sampling of reads. In our case, however, we have access to incomplete, fragmented assemblies instead of reads. Thus, to accurately use k-mer statistics, we first find orthologous regions by approximate alignment and then use k-mer statistics.

### Algorithm outline

The main idea behind skani is to find an approximate set of orthologous alignments between two genomes by obtaining a set of minimally overlapping k-mer chains[7] (the chains do not overlap much; the k-mers may overlap within the chain). We can then estimate ANI from the statistics of the k-mers in the chains, avoiding costly base-level alignment. The main algorithmic steps are listed below.

1. We use a very sparse set of *marker ℓ-mers* to estimate max-containment index and obtain a putative ANI using the FracMinHash method (section Sketching by Frac-MinHash). We filter out pairs of genomes with putative ANI <80% (section Max-containment putative ANI screening with marker ℓ-mers).
2. We select the genome with the larger score, defined as total sequence length times mean contig length, to be the reference and the other to be the query. We then fragment the query into 20-kb nonoverlapping chunks. In particular, this implies that the ANI computed by skani does not depend on the order of the inputs (that is, it is symmetric).
3. We extract $\frac{1}{c}$ fraction of k-mers for both genomes for some $c$ ($c = 125$ by default) as *seeds* to be used for chaining using FracMinHash (section Obtaining sparse seeds for chaining).

4. For each chunk on the query, we chain the seeds using a standard banded, heuristic chaining method against the reference (section Chaining sparse k-mer seeds).
5. We greedily extract minimally overlapping chains between the query and the reference and output aligned fraction (section Obtaining orthologous chains from homologous chains).
6. We estimate the ANI for each chunk, output the mean ANI over all chunks, and perform a learned ANI debiasing step (sections Estimating ANI from chains and Nonparametric regression for ANI debiasing).

### Sketching by FracMinHash

Instead of using the set of all k-mers in a genome, we use a compressed representation by sketching, by which we mean selecting only a subset of all k-mers. To obtain such a set of k-mers, we use the FracMinHash method[31]: given a hash function $h$ that maps k-mers to $[0, M]$, we select the k-mer $x$ as a seed if $h(x) < M/\gamma$, where $\gamma$ controls the fraction of selected k-mers. Assuming a uniform hash function, the expected fraction of selected k-mers is $\frac{1}{\gamma}$.

Although FastANI uses minimizer[32] k-mers to estimate Jaccard index and then ANI, recently, it was shown that Jaccard estimates (and thus ANI estimates) from minimizer k-mers are biased[27] and depend crucially on the window size $w$. Although this bias is not too bad when the $w$ is small (FastANI uses relatively small $w = 24$), it scales as $w$ increases. This means that a minimizer ANI estimator cannot use very sparse seeds, since the fraction of selected seeds is $\frac{2}{w+1}$ (ref. 33).

### Max-containment putative ANI screening with marker ℓ-mers

skani is not optimized for comparing distant genomes, so we can filter out comparisons against distant genomes using a very sparse set of FracMinHash ℓ-mers, which we call markers. We use these markers to estimate the max-containment index[14]; the same method is implemented in sourmash[25], although sourmash does not use max containment by default. Let a set of markers obtained from FracMinHash from the genome $G_1$ (with $\gamma = c_m$ and ℓ-mers) be denoted as $A$, and denote $B$ as the analogous set from the genome $G_2$. Assuming one of the genomes is contained in the other completely, we then calculate an ANI estimate between two genomes $G_1$, $G_2$ as

$$\text{ANI}_{FMH} = \left( \frac{|A \cap B|}{\min\{|A|, |B|\}} \right)^{1/\ell}.$$

The term on the right inside the exponent is the max-containment index. FracMinHash has only negligible bias in calculating the containment index[25] and can be used to obtain an estimate of ANI no matter the density (the fraction of selected k-mers). We let $\ell = 21$ for our marker ℓ-mers and $c_m = 1,000$ by default. We found skani's ANI algorithm was most accurate when ANI ≥ ~82%, so we only compare genomes with $\text{ANI}_{FMH} > 80\%$ as a conservative underestimate.

We can store all markers contained in the reference genomes as keys in a hash table with the associated values being the set of genomes containing the key (that is, an inverted index). Given a query genome, we can then obtain the intersection of all markers for the query against the references by checking all of its markers against the inverted index. This allows computation of the max-containment index in running time dependent on the number of similar genomes in the references[34]. From this, we can obtain $\text{ANI}_{FMH}$, making our filtering step fast as long as our reference and query genomes are diverse. When performing all-to-all pairwise comparisons, skani uses this strategy by default. However, when querying a small number of genomes against a database, skani iteratively checks all pairs of genomes by default because building the hash table takes a nontrivial amount of time.

## Obtaining sparse seeds for chaining

The marker $\ell$-mers described above are only used for filtering, but not for actually estimating the ANI. To actually estimate the ANI, we use a hybrid approach via local mapping (without base-level alignment) and containment index estimation. The first step in our approach is to obtain a different set of FracMinHash k-mers, this time taking $\gamma$, the sampling rate of k-mers, to be $\gamma = c$ where $c < c_m$, and $k < \ell$. This gives a denser, more sensitive set of k-mers to be used as *seeds*. These new k-mers are called seeds instead of markers because we actually use them as seeds for k-mer matching and alignment. We note that although we could have used other 'context-independent' k-mer seeding methods that are more 'conserved' than FracMinHash[35], we found that FracMin-Hash works well enough for relatively sparse seeds when $c \gg k$.

By default, $k = 15$ and $c = 125$. We note that the small value of $k = 15$ used by default leads to too many repetitive anchors on larger genomes, so we mask the top seeds that occur more than $2{,}500/c$ times by default. See the section Chaining score function and algorithm below for a justification of the choice of $2{,}500/c$.

## Choosing the $c$ parameter

The main parameter influencing runtime and accuracy is the $c$ parameter. The default value of $c = 125$ works well on a variety of tasks such as searching databases and MAG comparison. We found that for very fragmented and distant genomes, lowering $c$ may lead to more accurate ANI and AF estimates (Extended Data Figs. 5–7 and Supplementary Fig. 10). However, the runtime is inversely proportional to $c$. To guide users for choosing $c$, we suggest three different pre-set values of $c$ (in addition to the default) in skani v0.1.2's help messages based on empirical heuristics: a 'slow' pre-set with $c = 30$ for the most accurate AF estimates and pairs of genomes with $N50 \approx 3$ kb, a 'medium' pre-set with $c = 70$ for genomes with ANI ≤95 and $N50 \leq 10$ kb, and a 'fast' pre-set $c = 200$ for similar genomes with >95% ANI with $N50 > 10$ kb.

skani's index file sizes are inversely proportional to $c$, and a drawback of skani is that the index file sizes are larger than pure sketching methods (Supplementary Table 3), but the full indices can be stored on disk and only read into RAM by skani as needed (see skani implementation details).

## Chaining sparse k-mer seeds

After selecting one genome as a reference and the other as the query, we fragment the query into 20-kb nonoverlapping chunks. For each chunk, we find a set *anchors*, which are exact k-mer matches between $g$ and $g'$. Each anchor can be described by a tuple $(x, y)$, indicating the starting position of the matching k-mers on $g$ and $g'$ respectively. We collect the anchors into a strictly increasing subsequence called a *chain*[7] based on the ordering $(x_1, y_1) \prec (x_2, y_2)$ if $x_1 < x_2$ and $y_1 < y_2$.

We sort the anchors $(x_1, y_1), \ldots (x_N, y_N)$ in lexicographic order and let $S(i, j)$ be the score of chaining the $i$th anchor to the $j$th anchor. We wish to find a strictly increasing subsequence (based on our previously defined ordering $\prec$) of anchors $(i_1, i_2, \ldots)$ maximizing $\sum_\ell S(i_\ell, i_{\ell-1})$. The optimal such subsequence can be calculated in $O(N^2)$ time where $N$ is the number of anchors by the following dynamic programming: letting $f(i)$ be the optimal score of the chain up to the $i$th anchor, let

$$f(i) = \max\{\max_{i > j \geq 1} f(j) + S(i, j), 0\}.$$

After calculating $f(i)$ for each anchor $i$, we can obtain a set of optimal chains. We describe this in detail below.

## Chaining score function and algorithm

The chaining problem can be solved optimally in sub-quadratic time for a variety of chaining costs[8,9], but we opt for a simple heuristic method that is fast and good enough for our purposes instead. Letting $(x_i, y_i)$ and $(x_j, y_j)$ be the $i$th and $j$th anchors, we define our chaining cost to simply be $S(i, j) = 20 - |(y_j - y_i) - (x_j - x_i)|$. Notably, we allow and do not

penalize overlapping k-mers because this would bias the ANI calculation, as we want the chain to include all k-mers that arise from sequence homology. We also do not need to use more sophisticated scoring functions because we do not actually need to worry about base-level alignments or finding the longest chain. We use a banded dynamic programming method where instead of iterating over all $j < i$, we iterate only up to $i - A < j < i$ or stop if $|x_i - x_j| > B$ for some constants $A$ and $B$. Thus, the worst-case time is $O(AN)$ instead of $O(N^2)$. By default, we let $A = B/c$, where $B = 2{,}500$ and $c$ is the seed subsampling rate. This banded procedure is a simpler version of minimap2.23's chaining procedure, which also employs a heuristic to stop chaining early via the --max-chain-skip parameter.

## Obtaining homologous chains by backtracking

For each chunk, after computing all optimal scores $f(i)$ over all anchors using banded dynamic programming, we obtain optimal chains using the standard method of backtracking. That is, we store an array of pointers corresponding to anchors, where the pointer points to the optimal predecessor determined by the dynamic programming. For any anchor, we could then trace through this array to obtain the best chain corresponding to each anchor.

We wish to obtain a set of chains that do not share any anchors for each chunk. To do this, we partition the anchors into disjoint sets using a union-find data structure, taking unions of two anchor representatives whenever one anchor is an optimal predecessor of another. We then find the best $f(i)$ within each disjoint set and backtrack to obtain the optimal chain within each disjoint set. We take the set of all such chains over all chunks and call these chains homologous chains.

## Obtaining orthologous chains from homologous chains

It is possible that a single region chains to multiple paralogs, so the homologous chains obtained above may not be orthologous. We will use the term 'orthologous' loosely in the same sense as other ANI methods[6] – we denote a set of mappings (corresponding to chains) to be orthologous if they do not overlap too much along one of the genomes (that is, no duplicated one-to-all mappings). To obtain orthologous chains, we use a greedy minimally overlapping chain-finding procedure. We first sort every homologous chain (over all chunks) by its score and examine the best chain, only examining chains with three or more anchors. If the overlap between this current best chain and any other already selected chains is less than 50% of the current chain's length, we select this best homologous chain as an orthologous chain and then examine the next best homologous chain. We repeat this procedure until all remaining homologous chains overlap an orthologous chain by more than 50% and return all orthologous chains.

This procedure is similar in spirit to the reciprocal mapping method used in other ANI methods[6,19] to capture orthology, but it avoids performing alignment twice, making running time twice as fast. Other more sophisticated methods[36] exist for finding sets of orthologous alignments but we found this heuristic to be good enough for our method.

## Estimating ANI from chains

For a given chunk, let $\alpha$ be the number of anchors in the orthologous chains on that chunk, and $M$ be the number of seeds in the chunk. Then we estimate an ANI for each chunk by

$$\hat{\text{ANI}} = 1 - \hat{\theta} = \left(\frac{\alpha}{M}\right)^{1/k}.$$

This comes from modeling each k-mer as being an exact match and thus an anchor with probability $(1 - \theta)^k$ under a simple, independent mutation model, so the expected number of anchors is $(1 - \theta)^k$ times the number of k-mers[25].

However, the above formula fails when only part of the chunk is homologous to the reference, which may happen due to incompleteness of MAGs or structural variation. This will underestimate the ANI, since not all seeds in the chunk arise from sequence homology. Let $M_{LR}$ be the number of seeds in the chunk contained only between the leftmost and rightmost anchor over all minimally overlapping chains, and consider

$$\hat{ANI}_{LR} = \left( \frac{\alpha}{M_{LR}} \right)^{1/k}.$$

If $\hat{ANI}_{LR} > 0.950$ and the number of bases covered by in between these flanking anchors is >4*$c$, then we use $\hat{ANI}_{LR}$ as the estimate instead. This heuristic comes from the observation that when the ANI is large enough, our sparse chains are relatively accurate, so we can truncate the chunk with good accuracy. However, if ANI is small, it is more possible that this heuristic incorrectly excludes seeds that arise from sequence homology, so we avoid applying the heuristic to distant pairs of genomes.

### Estimating final ANI and AF

To obtain our final ANI estimates, we take a weighted mean over all chunks with weights given by $M$ (or $M_{LR}$ if the heuristic is applied instead), the number of seeds in the chunk. We also report the reference and query alignment fraction as the sum of the bases covered by all chains divided by the respective genome size. The number of bases in a chain is the last anchor position minus the first anchor position plus 2*$c$, where the 2*$c$ term is to account for k-mers near the edge of the chain missed by subsampling. By default, we only output a final ANI value if the alignment fraction is greater than 15%.

Because our final ANI estimate is just a weighted mean over the ANIs of the chunks, once we have these ANIs, we can quickly do bootstrapping by resampling the chunks and calculating the weighted mean over the resamples. We use 100 iterations and only proceed if there are >10 chunks, outputting the 5th and 95th percentile ANIs to be our final 90% confidence interval. We found that the bootstrap gave reasonable ANI confidence intervals over skani's inherent uncertainty (for example k-mer seeding variance), but we stress that it does not account for any systematic biases of skani's ANI estimator relative to other methods.

### Nonparametric regression for ANI debiasing

The chaining procedure can overestimate ANI, because the chains may exclude homologous but mutated k-mers near the edges of the chain due to the local mapping procedure. To handle this, we introduce an optional post-processing regression step to debias skani.

We trained a gradient boosted regression tree with absolute deviation loss where the target variable is ANIm's ANI calculation, and the features consist of the following: skani's ANI, standard deviation of skani's putative ANI distribution, and the 90th percentile contig lengths in the reference and query, and the average length of the k-mer chains, giving five features in total. We trained skani on a large, diverse set of 52,515 MAGs from Nayfach et al[2]. We computed all-to-all pairwise ANI values with >90% ANI according to an untrained version skani, and ran ANIm on the resulting 1,004,213 pairs of genomes. To tune hyperparameters such as tree depth, number of trees, and learning rate, we chose a set of human gut MAGs[37] that comes from a separate study than the training set and the datasets used in our results and then optimized our parameters over this new dataset.

To show that skani is not simply memorizing the organisms in the dataset, we partitioned this training set into two sets, $A$ and $B$, where each part is 'disjoint' from one another in the sense that skani does not output an ANI estimate between any MAG in $A$ and $B$ due to low aligned fraction (<15%). We show in Supplementary Figs. 6 and 11

that this debiasing procedure still corrects the ANI to be closer to a MUMmer-based ground truth.

We only debias comparisons with putative ANI > 90% and > 150, 000 aligned bases. We enable the debiasing procedure by default when the parameter $c$ is ≥70. We noticed that for smaller $c$, bias is less of an issue. We trained two models, one for $c = 125$ (default) and another for $c = 200$. If the user selects a specific value of $c$, the model with corresponding $c$ closer to the selected value is used. In particular, our results with $c = 125$ have debiasing enabled, but our results with $c = 30$ do not.

### skani implementation details

skani is implemented in the rust, a systems-level programming language, for speed. skani implements four primary subcommands: sketch, dist, search and triangle. The sketch command stores genomes in sketched representation (k-mer seeds and markers) for drop-in replacement for the other three commands. We use the same fast invertible k-mer hash function as minimap2 (ref. [38]) for sketching. Additionally, skani uses AVX2 SIMD instructions to do vectorized seeding of 64-bit k-mers in 256-bit lanes when AVX2 instructions are detected, which we found to speed up sketching by approximately 30%. Chaining was implemented naively with no hardware accelerations.

The three commands for sequence comparison, dist, triangle, and search, all calculate AF and ANI but have different runtime behavior. dist and triangle keeps all sketches in RAM, whereas the search command only keeps the marker k-mers in memory and allows for on-the-fly loading of the full sketch for each genome into RAM if the marker-based ANI threshold for the comparison is sufficiently high (≥ 80% by default), afterward discarding the index from RAM. This makes search much more memory efficient and possibly more time-efficient than dist when querying a large database, but it is IO-bound and not as efficient as dist when performing all-to-all comparisons. The triangle command is similar to dist but is limited to computing distance matrices, but because skani is symmetric, it performs only $n(n - 1)/2$ comparisons instead of all $n^2$. The inverted index for marker k-mer filtering is enabled by default in triangle, and otherwise in search and dist when the number of query genomes is ≥100. See Supplementary Fig. 12 for a benchmark on skani's runtime scaling with thread usage.

### Benchmarking details

All runtimes were benchmarked on a Intel Xeon CPU at 3.10 GHz machine with 64 cores and 240 GB RAM as a Google Cloud instance with a persistent SSD disk. Unless otherwise specified, all programs were run using 50 threads. Exact commands are shown in Supplementary Table 1.

### Reporting summary

Further information on research design is available in the Nature Portfolio Reporting Summary linked to this article.

## Data availability

All datasets are also specified in Supplementary Table 2 with descriptions. The Pasolli et al. 25-50 dataset[1] is available at http://segatalab.cibio.unitn.it/data/Pasolli_et_al.html. Ocean archaea MAGs[12] are available at https://doi.org/10.6084/m9.figshare.c.5564844.v1. Soil MAGs[17] are available at https://figshare.com/collections/Genomes_for_consistent_metagenome-derived_metrics_verify_and_define_bacterial_species_boundaries/4508162/1. Ocean eukaryotic MAGs[15,16] are available at https://osf.io/gm564/ and https://www.genoscope.cns.fr/tara/. Nayfach et al. MAGs[2] are available at https://genome.jgi.doe.gov/GEMs. *E. coli*, *B. anthracis*, and the D5 dataset genomes are available at http://enve-omics.ce.gatech.edu/data/fastani. GTDB-R207 (ref. [18]) database is available at https://gtdb.ecogenomic.org. *B. fragilis* genomes and the refseq-rc database is available at https://zenodo.org/record/8058221 (ref. [39]).

## Code availability

skani is available at https://github.com/bluenote-1577/skani. Scripts and notebooks for reproducing all figures is available at https://github.com/bluenote-1577/skani-test. skani v0.1.4's source code is deposited at https://zenodo.org/record/8058221 (ref. 39).

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

## Acknowledgements

J.S. was supported by an NSERC CGS-D scholarship. Work supported by Natural Sciences and Engineering Research Council of Canada (NSERC) grant RGPIN-2022-03074 and DND/NSERC Supplement DGDND-2022-03074.

## Author contributions

J.S. conceived the project, designed the algorithms and implemented skani. Y.W.Y. supervised and contributed to the development of the methods. Both authors wrote and edited the manuscript.

## Competing interests

The authors declare no competing interests.

## Additional information

**Extended data** is available for this paper at https://doi.org/10.1038/s41592-023-02018-3.

**Correspondence and requests for materials** should be addressed to Jim Shaw or Yun William Yu.

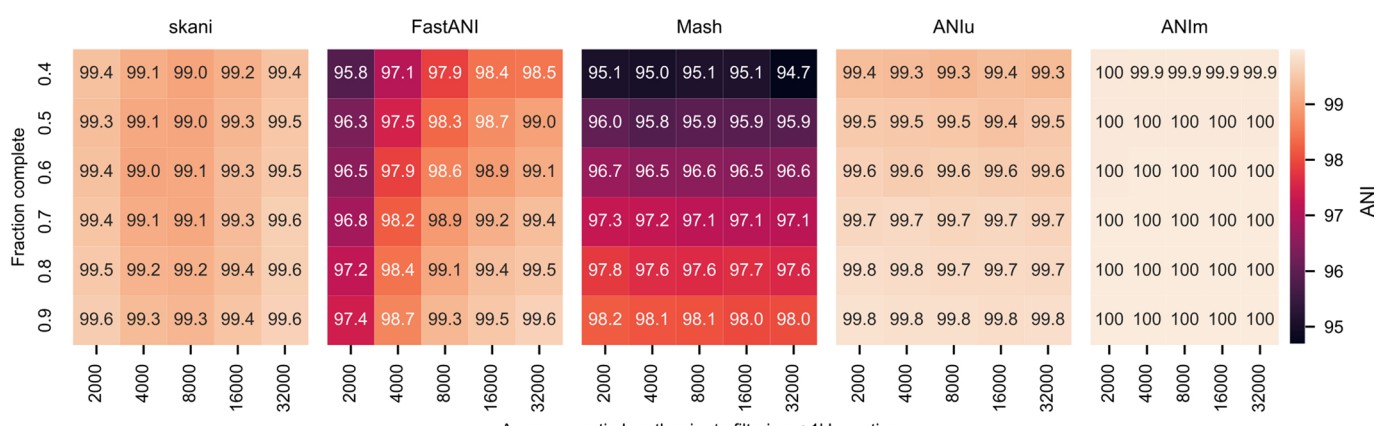

**Extended Data Fig. 1 | ANI benchmark under simulated fragmentation and incompleteness.** We fragmented an E. coli genome to obtain nonoverlapping contigs with lengths distributed according to an exponential distribution (mean length on the x-axis) and then retained each contig with some probability (probability on the y-axis) if the length was ≥ 1000 bp. These simulated MAGs (20 for each pair length and probability parameters) were compared to each other and the average is shown.

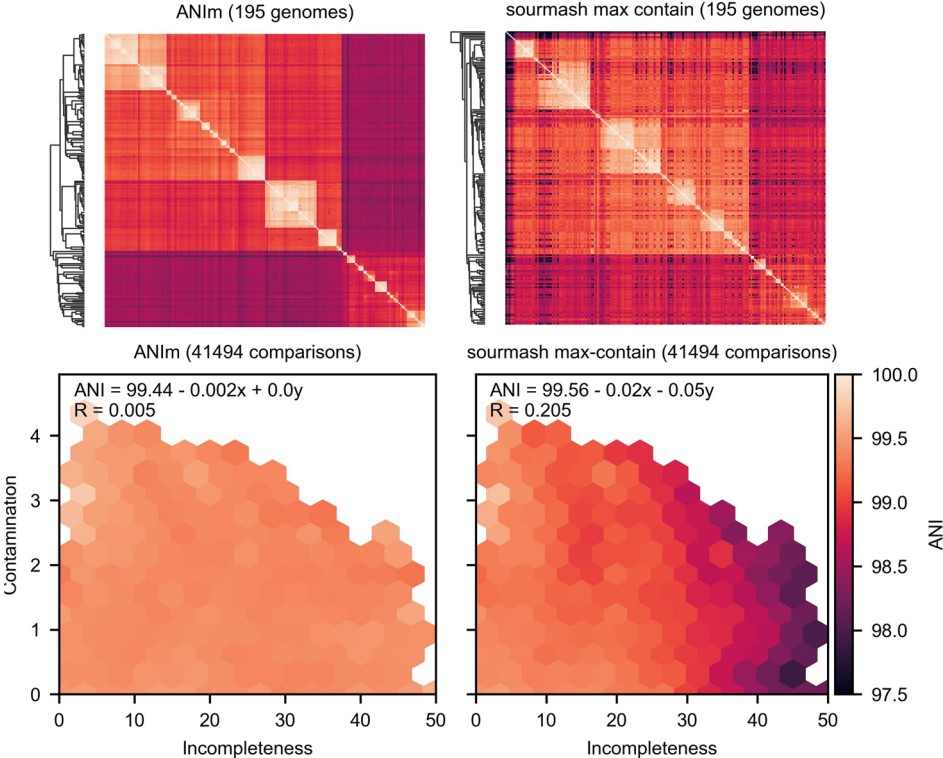

**Extended Data Fig. 2 | ANIm and sourmash experiments corresponding to the experiments shown in Fig. 1.** sourmash max-containment estimates ANI using the max-containment index, corresponding to the –max-containment option in sourmash.

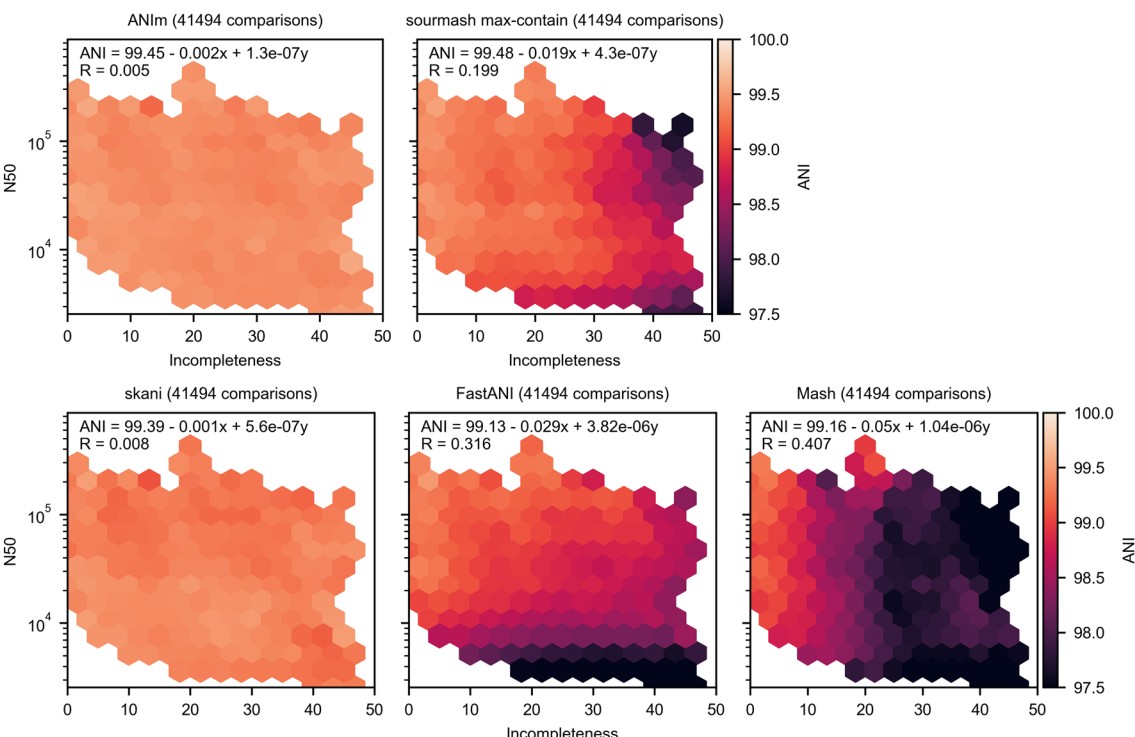

**Extended Data Fig. 3 | ANI sensitivity to fragmentation and incompleteness.** The same experiment was run as in Fig. 1b on Pasolli et al 25-50, except we used N50 as a covariate instead of contamination.

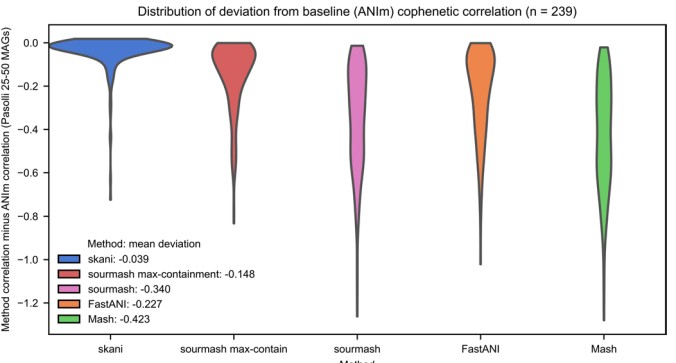

**Extended Data Fig. 4 | Violin plot of cophenetic correlation deviation for each method with respect to ANIm on the Pasolli 25-50 data set.** We took all species-level bins in the Pasolli data set with > 25 and < 50 genomes and computed distance matrices for all ANI methods. sourmash max-contain implies the --max-containment option in sourmash, otherwise sourmash was run with default parameters. For each bin, we compared the resulting clustering dendrograms for each method against ANIm's distance matrix by taking the cophenetic correlation. We plot the difference between this cophenetic correlation and the cophenetic correlation of ANIm's dendrogram to itself (which may be < 1 since ANIm's average-cluster dendrogram may not be perfectly concordant with itself). The legend shows the mean deviation of each method's cophenetic correlation from ANIm's cophenetic correlation.

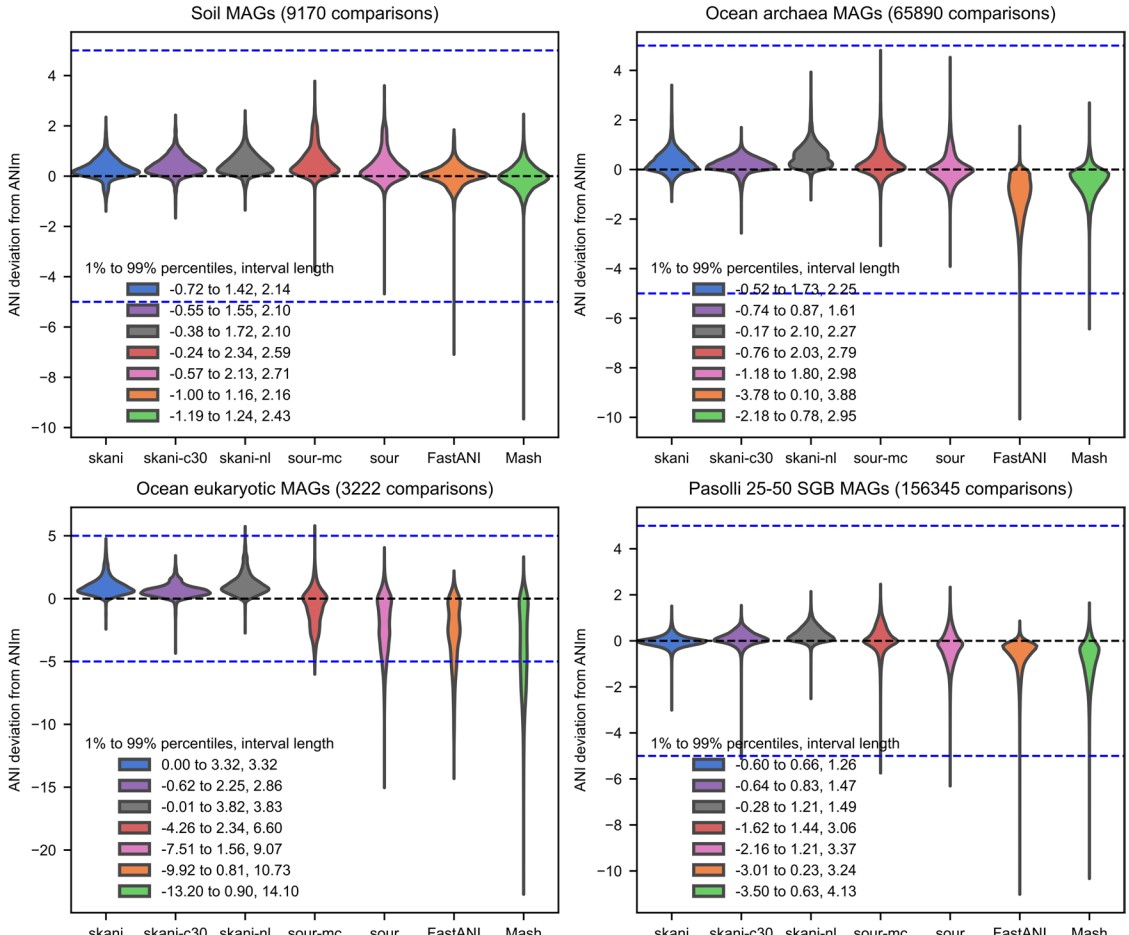

**Extended Data Fig. 5 | ANI deviation from ANIm for four data sets of MAGs with > 90% ANI (as predicted by ANIm).** See Supplementary Table 2 for data set descriptions. skani c-30 indicates skani with the parameter c set to 30, and skani-nl indicates no learned ANI regression debiasing (with default c = 125). sour-mc and sour correspond to sourmash with max-contain enabled and disabled. Blue lines show 5 and -5 deviation, and the legend indicates the 1 and 99 percentile deviations as well as their difference. The best 1% to 99% deviation distance for each data set is either skani or skani-c30. Lowering the c parameter generally improves results slightly, but is not guaranteed to do so; the default c = 125 gives results more in line with ANIm on the Pasolli 25-50 data set.

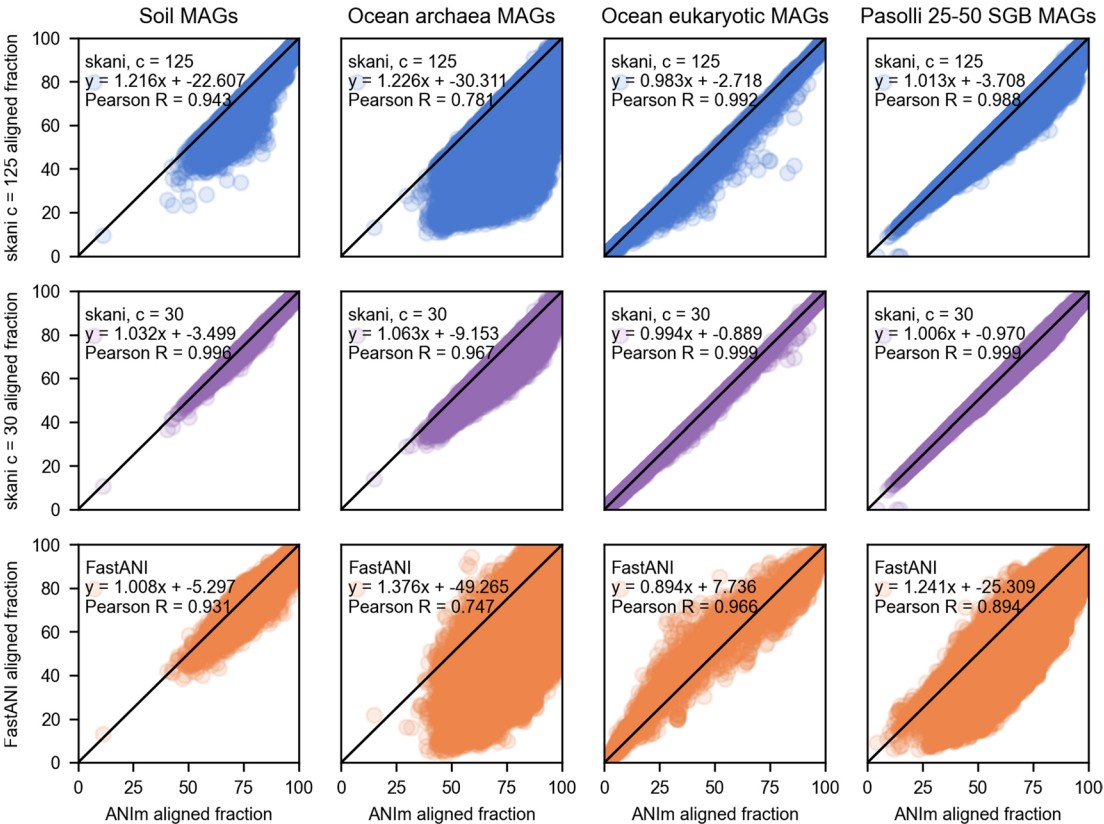

**Extended Data Fig. 6 | Aligned fraction correlation between ANIm and skani with default parameters (c = 125), skani with c = 30, and FastANI.** Only MAGs with > 90% ANI as predicted by ANIm are compared. Lowering the value of c for skani gives a more accurate signal. skani with default parameters still outperforms FastANI in terms of Pearson R for all data sets, and skani with c = 30 has near perfect correspondence with ANIm. The noisier results for archaea MAGs are likely due to its small N50 (median 5863 bp) and genome length (median 1.27 Mbp).

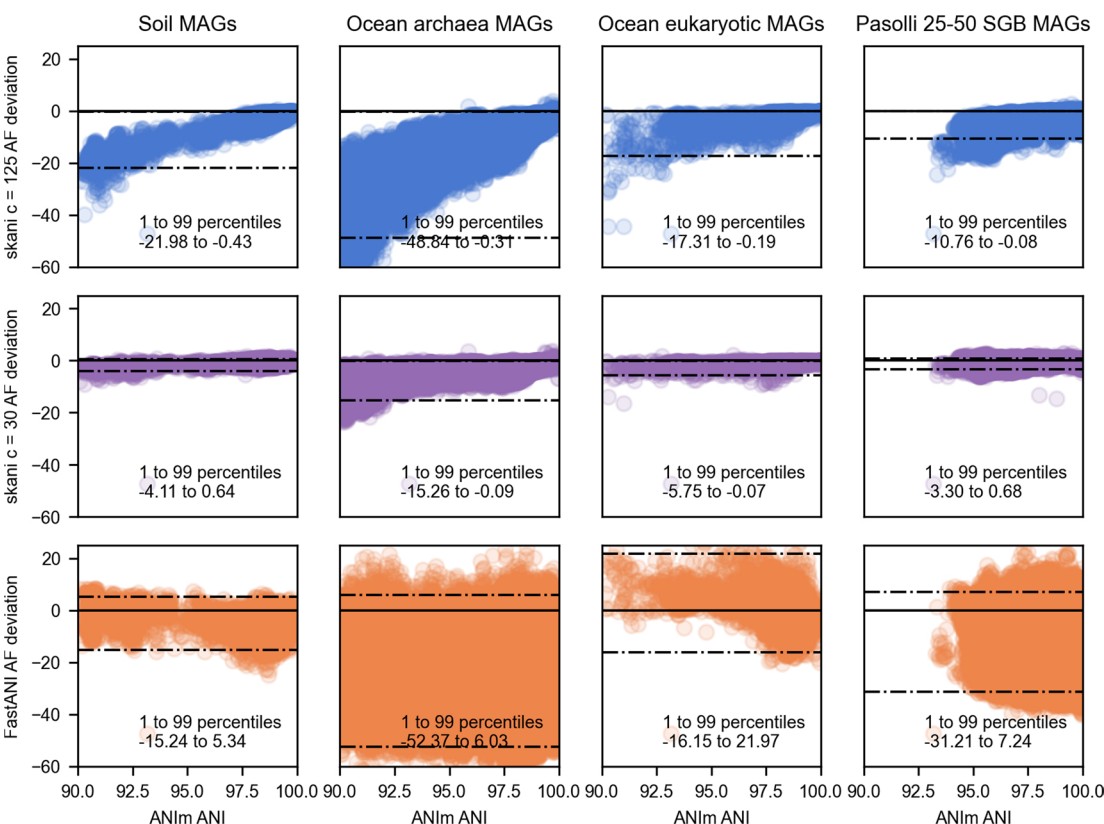

**Extended Data Fig. 7 | Aligned fraction deviation from ANIm as a function of ANI.** Dashed lines indicate the 1 and 99 percentiles of aligned fraction deviation from ANIm. Aligned fraction gets more accurate for skani as ANI increases. Decreasing c for skani improves the relationship between ANI and aligned fraction and decreases the variance as well.

# Reporting Summary

## Statistics

For all statistical analyses, confirm that the following items are present in the figure legend, table legend, main text, or Methods section.

| n/a | Confirmed | |
|---|---|---|
| ☐ | ☒ | The exact sample size (*n*) for each experimental group/condition, given as a discrete number and unit of measurement |
| ☐ | ☒ | A statement on whether measurements were taken from distinct samples or whether the same sample was measured repeatedly |
| ☒ | ☐ | The statistical test(s) used AND whether they are one- or two-sided<br>*Only common tests should be described solely by name; describe more complex techniques in the Methods section.* |
| ☒ | ☐ | A description of all covariates tested |
| ☐ | ☒ | A description of any assumptions or corrections, such as tests of normality and adjustment for multiple comparisons |
| ☐ | ☒ | A full description of the statistical parameters including central tendency (e.g. means) or other basic estimates (e.g. regression coefficient) AND variation (e.g. standard deviation) or associated estimates of uncertainty (e.g. confidence intervals) |
| ☒ | ☐ | For null hypothesis testing, the test statistic (e.g. *F*, *t*, *r*) with confidence intervals, effect sizes, degrees of freedom and *P* value noted<br>*Give P values as exact values whenever suitable.* |
| ☒ | ☐ | For Bayesian analysis, information on the choice of priors and Markov chain Monte Carlo settings |
| ☒ | ☐ | For hierarchical and complex designs, identification of the appropriate level for tests and full reporting of outcomes |
| ☐ | ☒ | Estimates of effect sizes (e.g. Cohen's *d*, Pearson's *r*), indicating how they were calculated |

*Our web collection on statistics for biologists contains articles on many of the points above.*

## Software and code

Policy information about availability of computer code

| Data collection | No software was used. |
|---|---|
| Data analysis | Our software skani v0.1.0 (described in manuscript and open-sourced at https://github.com/bluenote-1577/skani) was used for data analysis, along with the open source Python scikit-learn 0.22.1 package. Benchmarked against Mash v2.3, FastANI v1.33, sourmash v4.5, OrthoANIu (v1.2), ANIm pyani (v0.2.12). |

For manuscripts utilizing custom algorithms or software that are central to the research but not yet described in published literature, software must be made available to editors and reviewers. We strongly encourage code deposition in a community repository (e.g. GitHub). See the Nature Portfolio guidelines for submitting code & software for further information.

## Data

Policy information about availability of data

All manuscripts must include a data availability statement. This statement should provide the following information, where applicable:
- Accession codes, unique identifiers, or web links for publicly available datasets
- A description of any restrictions on data availability
- For clinical datasets or third party data, please ensure that the statement adheres to our policy

All data sets are specified in Supplementary Table 2 of the manuscript.

## Human research participants

Policy information about studies involving human research participants and Sex and Gender in Research.

| | |
|---|---|
| Reporting on sex and gender | N/A. (not human research) |
| Population characteristics | N/A. (not human research) |
| Recruitment | N/A. (not human research) |
| Ethics oversight | N/A. (not human research) |

Note that full information on the approval of the study protocol must also be provided in the manuscript.

## Field-specific reporting

Please select the one below that is the best fit for your research. If you are not sure, read the appropriate sections before making your selection.

☒ Life sciences ☐ Behavioural & social sciences ☐ Ecological, evolutionary & environmental sciences

For a reference copy of the document with all sections, see nature.com/documents/nr-reporting-summary-flat.pdf

## Life sciences study design

All studies must disclose on these points even when the disclosure is negative.

| | |
|---|---|
| Sample size | No sample-size calculation was performed. This is a software/benchmarking paper where we used fixed sample sizes to illustrate the relative performance against baselines of our and prior software. Sample sizes were chosen based on those used for previous studies and what was feasible for our computing platforms. |
| Data exclusions | No data were excluded. |
| Replication | As a methods and benchmarking paper, we were primarily interested in measuring the accuracy of out software. Thus, we repeated our analyses on multiple disparate data sets against gold-standard benchmarks. For data set generalizability, we ran on four different benchmark data sets and verified in the manuscript that the results were consistent.<br><br>Additioally, for software replicability, we ran our software on several different compute platforms, including a local machine, and on multiple Google Cloud instances. In every case, our software produced identical results. |
| Randomization | Not applicable, as we are designing a new ANI computation method, and all software was run on all datasets to compare. |
| Blinding | Not applicable, as this was a methods and benchmarking software design paper. |

## Reporting for specific materials, systems and methods

We require information from authors about some types of materials, experimental systems and methods used in many studies. Here, indicate whether each material, system or method listed is relevant to your study. If you are not sure if a list item applies to your research, read the appropriate section before selecting a response.

### Materials & experimental systems

| n/a | Involved in the study |
|---|---|
| ☒ ☐ | Antibodies |
| ☒ ☐ | Eukaryotic cell lines |
| ☒ ☐ | Palaeontology and archaeology |
| ☒ ☐ | Animals and other organisms |
| ☒ ☐ | Clinical data |
| ☒ ☐ | Dual use research of concern |

### Methods

| n/a | Involved in the study |
|---|---|
| ☒ ☐ | ChIP-seq |
| ☒ ☐ | Flow cytometry |
| ☒ ☐ | MRI-based neuroimaging |

