## [Peer Review File · Nature Methods]

Peer Review Information

Manuscript Title: Fast and robust metagenomic sequence comparison through sparse chaining with skani

Corresponding author name(s): Jim Shaw and Yun William Yu

Editorial Notes:

Reviewer Comments & Decisions:

Decision Letter, initial version:
--

28th Mar 2023

Dear Professor Yu,

Your Brief Communication, "Fast and robust metagenomic sequence comparison through sparse chaining with skani", has now been seen by 3 reviewers. As you will see from their comments below, although the reviewers find your work of considerable potential interest, they have raised a number of concerns. We are interested in the possibility of publishing your paper in Nature Methods, but would like to consider your response to these concerns before we reach a final decision on publication.

We therefore invite you to revise your manuscript to fully address these concerns.

* include a point-by-point response to the reviewers and to any editorial suggestions

* please underline/highlight any additions to the text or areas with other significant changes to facilitate review of the revised manuscript

- * address the points listed described below to conform to our open science requirements
- * ensure it complies with our general format requirements as set out in our guide to authors at www.nature.com/naturemethods
- * resubmit all the necessary files electronically by using the link below to access your home page

[REDACTED]

We hope to receive your revised paper within 3 months. If you cannot send it within this time, please let us know. In this event, we will still be happy to reconsider your paper at a later date so long as nothing similar has been accepted for publication at Nature Methods or published elsewhere.

OPEN SCIENCE REQUIREMENTS

REPORTING SUMMARY AND EDITORIAL POLICY CHECKLISTS

DATA AVAILABILITY

We strongly encourage you to deposit all new data associated with the paper in a persistent repository where they can be freely and enduringly accessed. We recommend submitting the data to discipline-specific and community-recognized repositories; a list of repositories is provided here:

<http://www.nature.com/sdata/policies/repositories>

All novel DNA and RNA sequencing data, protein sequences, genetic polymorphisms, linked genotype and phenotype data, gene expression data, macromolecular structures, and proteomics data must be deposited in a publicly accessible database, and accession codes and associated hyperlinks must be provided in the "Data Availability" section.

CODE AVAILABILITY

Please include a "Code Availability" subsection in the Online Methods which details how your custom code is made available. Only in rare cases (where code is not central to the main conclusions of the paper) is the statement "available upon request" allowed (and reasons should be specified).

For more information on our code sharing policy and requirements, please see: <https://www.nature.com/nature-research/editorial-policies/reporting-standards#availability-of-computer-code>

MATERIALS AVAILABILITY

Authors reporting new chemical compounds must provide chemical structure, synthesis and

characterization details. Authors reporting mutant strains and cell lines are strongly encouraged to use established public repositories.

ORCID

Sincerely,

Lin Tang, PhD
Senior Editor
Nature Methods

Reviewers' Comments:

Reviewer #1:

Remarks to the Author:

Shaw and colleagues describe a fast heuristic method for determining the ANI and AF between genomes. The proposed method is shown to compare favourably to existing heuristics such as FastANI and Mash, especially on lower-quality genome assemblies as is typical for MAGs. The results are close enough to those obtained using more computationally expensive, "gold standard" methods such as ANIm to be of practical use in many applications. I was able to install and run skani without issue. I found the provided results to be thorough and convincing, and thank the authors for their efforts in developing this software package. I have a few suggestions that I believe will help clarify a few results.

Major:

- It is challenging to qualitatively compare the dendrograms produced by the different methods given their size and lack of extent labels. The cophenetic correlation results are much appreciated, but indirectly speak to the similarity of the resulting average-linkage trees. Can the similarity between trees be quantified using a tree comparison measure such as Robinson-Foulds?

- Most results use ANIm as the "gold standard". However, Fig. 2 and Extended Data Fig. 7 use OrthoANIu. This inconsistency is a bit confusing especially since Extended Data Fig. 2 justifies the use of ANIm as the superior method. I think many consider ANIu to be a heuristic of ANIb (BLAST-based ANI) since it relies on USEARCH so lacks the historical importance of ANIb. Can all results be made relative to ANIm for consistency and because this is shown to be the most principled method?

Minor:

- Extended Data Fig. 2 is referenced before Extended Data Fig. 1; similarly, Extended Data Figs. 11-14 referenced before Figs. 7-10.

- Caption for Extended Data Fig. 11 has an upside down question mark that I believe should be a greater than sign.

- Extended Data Fig. 11 would be easier to interpret with a horizontal line designating a deviation of 0.

- Does the "-n <n>" parameter ensure the highest n reference genomes in terms of ANI?

- skani triangle outputs a lower-triangular matrix. Based on this and the method description, my understanding is that skani is symmetric in terms of ANI (i.e. same results regardless of what genome is the reference or query)? I'd recommend emphasizing this in the documentation and potentially the manuscript text since many ANI methods are not symmetric. In practice, this often means calculating ANI twice for a pair of genomes.

Reviewer #2:

Remarks to the Author:

Summary of the key results

In this manuscript, Shaw and Yu introduce skani. Skani is a novel method for computing the average nucleotide identity (ANI) between pairs of sequences — most commonly between assembled or partially assembled genomes. The authors claim that skani retains the speed of sketching based methods, like MASH and sourmash, while simultaneously providing much more accurate results in the (realistic) context of highly-fragmented and contaminated metagenome assembled genomes (MAGs).

Therefore, the high-level contributions are (1) a new method that combines fast filtering via "marker l-mer" sketches, (2) an efficient seeding and co-linear chaining approach on references that pass the filter, (3) intelligent (greedy) extraction of minimally-overlapping chains (4) some modified / optimized definitions used to compute ANI from these chains and (5) optionally, an ability to further debias estimates using a learned regression model. Additionally, the authors provide an efficient implementation of this algorithm in rust that can be built and installed using the standard rust toolchain (and which they have also made available via bioconda).

The improved results are demonstrated using both simulated data where the true ANI is known, as well as comparison on several experimental datasets with other "gold standard" ANI computation methods whose results are trustworthy but which are typically too slow to use in practice on a large scale.

Originality and significance: if not novel, please include reference

While the fundamental building blocks used to compose skani (for the most part) are not original, the careful construction of the method itself to produce an approach and tool for both fast and accurate ANI calculation is original.

Specifically, most of the key components exist elsewhere in the broader literature (e.g. methods like sourmash allow using fracminhash to estimate ANI, co-linear chaining has been used in read alignment methods since BWA and the approach in skani is seemingly similar to that used in minimap2, likewise with the extraction of non-overlapping / minimally-overlapping chains).

However, the careful piecing together of these distinct methods — the joining of sparse fracminhash filtering with sensitive chaining and greedy chain extraction — is novel. Further, certain components of the approach do appear novel, like the modified ANI equations used to account for seeds missed near the end of chains (i.e. $\hat{\mathrm{ANI}}_{\{LR\}}$) and the regression-based approach to additional debiasing of ANI estimates.

Personally, I judge the significance of the work to be substantial. I say this because (a) the problem being addressed is one of substantial interest, as evidenced by the popularity of the tools that are being compared with (b) the argument that the scale of the data necessitates computationally efficient solutions is supported, practically, by the popularity of sketching methods (e.g. Mash and sourmash) and hybrid methods (e.g. FastANI) and (c) the practical demonstration in the bias introduced in the ANI calculation performed by existing tools — following up on the theoretical and practical work of Belbasi et al. — is well demonstrated herein.

In response to these challenges, the authors provide their own hybrid approach that largely overcomes the bias problems exhibited by other approaches while remaining very fast (though, unsurprisingly, still slower than purely sketching-based approaches in some contexts). Moreover, and this seems critical, they have produced a seemingly solid piece of software implementing their approach that is easy to install and use. Ultimately, their dedication to continuing to support and improve the tool will impact its level of significance, but the current offering is impressive.

Data & methodology: validity of approach, quality of data, quality of presentation

The approach is intuitive, as is the explanation of how it works. The evaluations are made across a broad range of different data sets (with different characteristics), which build confidence that the method produces robust results that compare well with those of existing methods.

The presentation is clear and the method is relatively straightforward to understand. I do think that the current manuscript would benefit from some extra detail (in the Extended Data or Online Methods) describing important implementation details (i.e. what care had to be taken to make the method as fast as it is) and conveying what the current bottleneck is (i.e. is it computational, I/O, etc.).

Appropriate use of statistics and treatment of uncertainties

The method presented in this manuscript is deterministic. The relevant analyses seem appropriate.

Conclusions: robustness, validity, reliability

The conclusions drawn in the manuscript seem appropriate. Testing is performed across a variety of different datasets and across datasets conveying different characteristics. Thus, it appears these results are robust across a range of different settings and that the method generally performs well in terms of accuracy, runtime, and memory requirements.

Suggested improvements: experiments, data for possible revision

* All evaluations in the manuscript were carried out with 50 threads. While this is not an unreasonable number, and one might expect many users to run skani and other programs on large servers offering many concurrent threads of execution, I believe the manuscript would benefit from a scalability analysis on at least one of the large datasets being evaluated. How does runtime vary with the number of threads used? Does one expect that all of the methods would continue to scale well if more threads were available, or have some of them already saturated?

* From comparing the wall clock times to the CPU times, it seems that e.g. Mash is able to make better use of many (50) threads. Is there a reason that parallelization is easier there, or is this just an artifact of the current implementation. Also, it would be useful to explore and describe what the relevant bottleneck step(s) currently are in skani. For example, is skani CPU limited during the chaining and chain extraction steps, or does it simply take longer to deserialize from the (larger) on-disk databases. If the latter is the case (i.e. if CPU utilization is substantially below all 50 threads), then perhaps it might make sense to trade some computation to reduce I/O overhead by, e.g., block-compressing the database on disk.

* Likewise, the parameter c (the inverse of the fraction of selected seed k -mers) seems a critical one in skani. For example, the authors demonstrate (in Extended Data Figure 12) that in particularly challenging cases, where the default value of 125 struggles, skani with $c=30$ does very well. Is there some way to assess or determine when a particular value of c should be changed? On one hand, the larger c , the sparser the chaining and the faster it seems the algorithm will run. On the other hand, in more fragmented MAGs, a denser sampling can seemingly be more sensitive and lead to better ANI estimates. When should users consider changing the default? Is there any diagnostic that can be used to inform the user that the current value of c may be inappropriate, or at least a rule-of-thumb that can be used to select a reasonable c ?

References: appropriate credit to previous work?

Overall, the authors do a good job, in my opinion, of including appropriate references and citing them in the proper context.

One place where I think relevant references could (and should) be added is in the discussion of the chaining, chain scoring, and chain extraction sections. Specifically, the co-linear chaining algorithm introduced here is a standard approach. While Abouelhoda and Ohlebusch are appropriately cited, no citation is given for the "banded dynamic programming" heuristic used in skani. However, both the co-linear chaining formulation and the heuristic to speed up the co-linear chaining are very similar to what is used in minimap2 (Li 2018). There, chaining is accelerated by limiting the number of predecessors that will be considered for extending a chain given the intuition that, since predecessors are traversed in reverse order, from closest to farthest, once a good predecessor is found off of which

to extend the current chain, it is unlikely that more distant predecessors will produce better scores. A similar relationship, though less direct, seems to hold between the approach used to extract minimally-overlapping chains in skani and the approach taken for a similar purpose in minimap2.

Finally, given the centrality of co-linear chaining to the skani approach, it seems worth to mention recent work on the co-linear chaining problem, like "Chaining with overlaps revisited" (Mäkinen and Sahlin 2020: <https://arxiv.org/abs/2001.06864v2>) and "Co-linear Chaining with Overlaps and Gap Costs" (Jain et al. RECOMB 2022). These describe increasingly practical $\tilde{O}(n)$ (where $\tilde{O}(\cdot)$ hides polylog(n) factors) co-linear chaining algorithms, and the latter paper also demonstrates important relationships between the edit distance and optimal co-linear chaining score that seem relevant, at least in theory, to the use herein of orthologous chains to estimate the ANI.

Clarity and context: lucidity of abstract/summary, appropriateness of abstract, introduction and conclusions

The abstract and conclusion are, in my opinion, clear and appropriate. The introduction describing and categorizing related work is brief, but appropriate in the context of the article type being submitted.

Other minor comments / typos

- * Extended figure 2 should show the color bar legend.
- * Extended Data figure 11 has a math mode escape issue in the caption.

Reviewer #3:

Remarks to the Author:

Shaw and Yu describe a tool called skani for accurate estimation of the ANI between two MAGs despite incompleteness and contamination. The selling points of the tool are its speed and its accuracy in the face of incompleteness/contamination. The accurate calculation of ANI is a topic of great interest. And results were convincing to me that the method is able to compute ANI more accurately than fast methods and faster than accurate methods. Thus, I find the work to be valuable overall. Below are some comments to improve the manuscript.

Major

1. There must be a limit at which any method of ANI calculation will start to fail due to incompleteness and/or contamination. Reading the paper, I didn't get any sense of what that level is for skani. I would like to see an experiment that tells me skani can still compute accurate distances as long as incompleteness is smaller than X or contamination is smaller than Y. Or something along those lines. Basically, I'd like to see the limits of the method in terms of input data quality. This analysis will be extremely helpful to readers and will save authors the headache in the future to explain to naive users that such and such datasets should not have been used with this method, to begin with (e.g., two completely non-overlapping contigs of the same genome).
2. It does not come through in the abstract and much of the intro that the focus is on very high ANI. A lot of us find low ANI values also interesting. This comes through later but should be clarified in the abstract and/or intro.

3. While most of the methodological steps were based on concepts I could easily accept, the idea that the under-estimation bias is solved using a trained model makes me somewhat worried, especially because that's the most interesting challenge in terms of accuracy. I ask for one clarification for line 291 below in minor comments. And I did note that the authors have only a small number of features, which is good. But, several other analyses would help me become more confident:

* Show clearly the comparison between skani with and without this correction.

* Show analyses where the bias correction model is trained on a dataset where *every* training MAG has low (e.g., < 80) ANI to *every* MAG in the testing set (or just train on archaea and apply on bacteria). Does the bias correction still work? If yes, I will feel very confident (I am not convinced by the vague "disjoint" assertion). If not, perhaps the method will not scale quite as well to entirely novel groups.

4. Authors can do a better job of situating the work within the literature. They do a good job of covering methods focused on MAG ANI. But there is a distinct but related literature on ANI calculation from short reads. In particular, in that literature, the idea that incompleteness can mislead Jaccard was noted and studied in these previous studies (at least three, shown below). It would be necessary to discuss them. In particular, Afann uses machine learning to correct for low coverage in ways that may be related to the proposed methods. Skmer, in contrast, uses a formula, which is quite different from what is done here. I am not suggesting that an actual comparison to these methods is necessary (since they are designed for reads). But the conceptual connection should be made.

* Tang, Kujin, Jie Ren, and Fengzhu Sun. "Afann: Bias Adjustment for Alignment-Free Sequence Comparison Based on Sequencing Data Using Neural Network Regression." *Genome Biology* 20, no. 1 (December 4, 2019): 266. <https://doi.org/10.1186/s13059-019-1872-3>.

* Sarmashghi, Shahab, Kristine Bohmann, M. Thomas P. Gilbert, Vineet Bafna, and Siavash Mirarab. "Skmer: Assembly-Free and Alignment-Free Sample Identification Using Genome Skims." *Genome Biology* 20, no. 1 (December 13, 2019): 34. <https://doi.org/10.1186/s13059-019-1632-4>.

* Fan, Huan, Anthony R. Ives, Yann Surget-Groba, and Charles H. Cannon. "An Assembly and Alignment-Free Method of Phylogeny Reconstruction from next-Generation Sequencing Data." *BMC Genomics* 16, no. 1 (December 14, 2015): 522. <https://doi.org/10.1186/s12864-015-1647-5>.

5. The simulation study was a bit too sparse in my opinion (real data analyses were great). Why only test 100% true ANI? Why not also throw mutations at random on a base genome with a known theta so that you can measure accuracy? Also, contamination effects were not studied. You can combine in some random contig from an unrelated genome and create a chimera.

Minor

* Figure SE: 5% error does not seem bad to readers who don't understand the applications of ANI. You should perhaps emphasize why 5% is such a higher error for ANI=100%. Describe applications where that error is catastrophic.

* Lines 74-79. The authors say "phylogenetic observations" (which is somewhat oddly worded). But then, they use a non-phylogenetic method (average linkage clustering). Why not use something more phylogenetic like NJ or FASTME. It doesn't matter for these low distances but you may also want to do JC correction. This point is not crucial due to the low distances.

- * Line 108: constant time with respect to what? The number of reference genomes? Something else? I am not even sure if this is true after reading the Methods. Make the claim more precise or remove it.
- * Line 127: The sentence reads too vague.
- * Line 147: Not clear what you mean but "such spurious matches". Seems like a sentence is left out.
- * Line 164: Would be good to give some idea here of how these k-mers are selected. Perhaps name the method (min hash)?
- * Line 159-170: Hard to connect steps 1-6 to the subsequent subheadings. Use keywords in steps or use step numbers in the heading to make it easy to connect.
- * Equation after line 189: I don't get why this is correct unless the part of the MAG covered by A is a subset of the parts covered by B (or vice versa). Or is this equation written only for the case where A and B are both complete genomes?
- * Truly sub-linear is too strong here. If I understand correctly, you compute the intersection with any genome that shares at least one marker kmer with the query. If not, please specify how you decide which references to compare against (a minimum number of markers needed)? If you do that, then, all it takes to increase the number of reference genomes is one "bad" marker (e.g., one that is ubiquitous). Since MinHash does not take into account things like the frequency and complexity of a kmer, there is no guard against this. I think, at a minimum, low complexity kmers should be disqualified as markers.
- * Lines 249-255: Calling these orthologous is a stretch. If all goes well, this is just a closest-match procedure, which is not the same as orthology. I agree the distinction doesn't matter for you but why call it orthology and confuse readers/users?
- * Line 277: I am not sure what a percentile bootstrap is and how it differs from normal bootstrap. More broadly, I don't understand what object you are subsampling *with replacement*. What is the meaning of having sampled the same chunk twice? Perhaps this is not bootstrapping and just subsampling?
- * Line 291: Related to major comment 3: What does "disjoint" mean? Do they have high ANI to all the MAGs used for testing? Or just that it's a different dataset? In terms of "overfitting" (the subcontext of this paragraph), it doesn't matter whether the MAG is called the same or not if a MAG in the testing has low ANI to a MAG in training.
- * Latex error: $\hat{\epsilon}$ 90% caption of Figure E11. Also, adding a horizontal line at zero helps the eye here.

Author Rebuttal to Initial Comments

Point-by-point reviewer response.

Response legend:

Blue - Reviewer

Black - Authors.

Red - New additions to the text. Also outlined in red in the revised manuscript.

Reviewer response 1:

Remarks to the Author:

Shaw and colleagues describe a fast heuristic method for determining the ANI and AF between genomes. The proposed method is shown to compare favourably to existing heuristics such as FastANI and Mash, especially on lower-quality genome assemblies as is typical for MAGs. The results are close enough to those obtained using more computationally expensive, “gold standard” methods such as ANIm to be of practical use in many applications. I was able to install and run skani without issue. I found the provided results to be thorough and convincing, and thank the authors for their efforts in developing this software package. I have a few suggestions that I believe will help clarify a few results.

R1.1. We thank the reviewer for their positive feedback on our results and our software.

Major:

- It is challenging to qualitatively compare the dendrograms produced by the different methods given their size and lack of extent labels. The cophenetic correlation results are much appreciated, but indirectly speak to the similarity of the resulting average-linkage trees. Can the similarity between trees be quantified using a tree comparison measure such as Robinson-Foulds?

R1.2. We tried the reviewer’s suggestion and applied Robinson-Foulds to average-linkage trees in Fig. 1 against ANIm’s tree. The RF distance between skani, mash, and FastANI against ANIm’s tree was 0.489, 0.713, and 0.823, which is in general concordance with the cophenetic correlation values (0.968, 0.718,

and 0.507 cophenetic correlations, measuring similarity instead of distance). We add the following text to the caption of Fig. 1:

Fig. 1 caption:

The Robinson-Foulds distances for skani, FastANI, and Mash's dendrograms against ANIm's average-linkage tree are 0.489, 0.713, and 0.823 respectively.

We hypothesize that the reason the RF distance between skani and ANIm's tree is still quite high (0.489) is because we are comparing extremely similar strains with small ANI differences. For example, three strains can have pairwise ANIs 99.9, 99.8, 99.85, leading to a tree with small branch lengths. Because the standard RF distance **only takes into account topology and not branch lengths**, it is highly sensitive to these small differences. Cophenetic correlation, however, takes into account branch lengths.

While there are weighted versions of Robinson-Foulds, the unweighted versions appear to be more popular in practice. We also found a lack of weighted Robinson-Foulds distance implementations, with not much documentation available. As such, we have decided to keep our measurements based on cophenetic correlation while adding a note in the text regarding tree-distance usage:

Line 86:

Importantly, cophenetic correlation takes into account branch lengths in the dendrogram, unlike the commonly used unweighted Robinson-Foulds distance [17] between trees.

- Most results use ANIm as the "gold standard". However, Fig. 2 and Extended Data Fig. 7 use OrthoANlu. This inconsistency is a bit confusing especially since Extended Data Fig. 2 justifies the use of ANIm as the superior method. I think many consider ANlu to be a heuristic of ANIb (BLAST-based ANI) since it relies on USEARCH so lacks the historical importance of ANIb. Can all results be made relative to ANIm for consistency and because this is shown to be the most principled method?

R1.3. We agree that we were not clear on why we used (Ortho)ANlu instead of ANIm in Fig. 2 and thank the referee for this comment.

The reason we used ANI_u in Fig. 2 is because ANI_m is not accurate with computing ANI for pairs of genomes with < 90% ANI. This is also noticed in the article “A large-scale evaluation of algorithms to calculate average nucleotide identity” by Yoon et al (2017). For the GTDB dataset in Fig. 2, we tried using ANI_m and got the following relationship between ANI_m and ANI_u:

The low clump of ANI comparisons around 82% ANI is very discordant between the ANI_m and ANI_u, and in fact, skani is better than ANI on this dataset in this regime. Because we wanted to showcase skani below 90% ANI as well, we decided to use ANI_u in Fig. 2.

This is why when using ANI_m as a baseline (in Fig.1 or extended data figures), we only ever used it for comparing ANI > 90%.

We agree that ANlu lacks the historical importance of ANIb. However, USEARCH, which ANlu is based on, is used quite often in practice (> 18000 citations currently) by practitioners due to its speedup over blast, so benchmarks may be of use to practitioners. ANlu is also shown to be very comparable to ANIb in the aforementioned Yoon et al paper, as well as in “All ANIs are not created equal: implications for prokaryotic species boundaries and integration of ANIs into polyphasic taxonomy” Fig. 1 by Palmer et al (> 0.998 R² value correlation in both studies).

To clarify our usage of ANlu, added the following justification for our use of ANlu:

Line 134:

We benchmarked skani against OrthoANlu [22,23] (which we shorten to ANlu), a faster but almost-identical analogue of the BLAST-based ANIb, as a baseline. We did not use ANIm because ANIm overestimates ANI for pairs of genomes with < 90% ANI [24,23].

Minor:

- Extended Data Fig. 2 is referenced before Extended Data Fig. 1; similarly, Extended Data Figs. 11-14 referenced before Figs. 7-10.

R1.4. Fixed.

- Caption for Extended Data Fig. 11 has an upside down question mark that I believe should be a greater than sign.

R1.5. Fixed.

- Extended Data Fig. 11 would be easier to interpret with a horizontal line designating a deviation of 0.

R1.6. We have added a black dashed line indicating 0 deviation (now Extended Data Fig. 10).

- Does the “-n <n>” parameter ensure the highest n reference genomes in terms of ANI?

R1.7. Yes, this is the right interpretation of the -n parameter. We’ve changed the help message to be more clear in skani v0.1.2.

- skani triangle outputs a lower-triangular matrix. Based on this and the method description, my understanding is that skani is symmetric in terms of ANI (i.e. same results regardless of what genome is the reference or query)? I’d recommend emphasizing this in the documentation and potentially the manuscript text since many ANI methods are not symmetric. In practice, this often means calculating ANI twice for a pair of genomes.

R1.8. The referee is correct in that skani is symmetric in terms of ANI. This is because in step 2. of subsection “Algorithm outline” in Methods, we choose one genome to be a reference based on the sequence lengths and mean contig sizes of the two genomes, so it is independent of the input order. We have added the following sentence to the methods section:

Line 180:

In particular, this implies that the ANI computed by skani does not depend on the order of the inputs (i.e. it is symmetric).

Furthermore, we have changed the documentation in the github to make it more clear it is symmetric in the “Quick start” section of the README.

Reviewer response 2:

Remarks to the Author:

Summary of the key results

In this manuscript, Shaw and Yu introduce skani. Skani is a novel method for computing the average nucleotide identity (ANI) between pairs of sequences — most commonly between assembled or partially assembled genomes. The authors claim that skani retains the speed of sketching based methods, like MASH and sourmash, while simultaneously providing much more accurate results in the (realistic) context of highly-fragmented and contaminated metagenome assembled genomes (MAGs).

Therefore, the high-level contributions are (1) a new method that combines fast filtering via "marker l-mer" sketches, (2) an efficient seeding and co-linear chaining approach on references that pass the filter, (3) intelligent (greedy) extraction of minimally-overlapping chains (4) some modified / optimized definitions used to compute ANI from these chains and (5) optionally, an ability to further debias estimates using a learned regression model. Additionally, the authors provide an efficient implementation of this algorithm in rust that can be built and installed using the standard rust toolchain (and which they have also made available via bioconda).

The improved results are demonstrated using both simulated data where the true ANI is known, as well as comparison on several experimental datasets with other "gold standard" ANI computation methods whose results are trustworthy but which are typically too slow to use in practice on a large scale.

Originality and significance: if not novel, please include reference

While the fundamental building blocks used to compose skani (for the most part) are not original, the careful construction of the method itself to produce an approach and tool for both fast and accurate ANI calculation is original.

Specifically, most of the key components exist elsewhere in the broader literature (e.g. methods like sourmash allow using fracminhash to estimate ANI, co-linear chaining has been used in read alignment methods since BWA and the approach in skani is seemingly similar to that used in minimap2, likewise with the extraction of non-overlapping / minimally-overlapping chains).

However, the careful piecing together of these distinct methods — the joining of sparse fracminhash filtering with sensitive chaining and greedy chain extraction — is novel. Further, certain components of

the approach do appear novel, like the modified ANI equations used to account for seeds missed near the end of chains (i.e. $\hat{\mathit{ANI}}_{LR}$) and the regression-based approach to additional debiasing of ANI estimates.

Personally, I judge the significance of the work to be substantial. I say this because (a) the problem being addressed is one of substantial interest, as evidenced by the popularity of the tools that are being compared with (b) the argument that the scale of the data necessitates computationally efficient solutions is supported, practically, by the popularity of sketching methods (e.g. Mash and sourmash) and hybrid methods (e.g. FastANI) and (c) the practical demonstration in the bias introduced in the ANI calculation performed by existing tools — following up on the theoretical and practical work of Belbasi et al. — is well demonstrated herein.

In response to these challenges, the authors provide their own hybrid approach that largely overcomes the bias problems exhibited by other approaches while remaining very fast (though, unsurprisingly, still slower than purely sketching-based approaches in some contexts). Moreover, and this seems critical, they have produced a seemingly solid piece of software implementing their approach that is easy to install and use. Ultimately, their dedication to continuing to support and improve the tool will impact its level of significance, but the current offering is impressive.

R2.1. We thank the reviewer for highlighting 1) the novelty of our algorithm, which stems from piecing together different parts of existing algorithms with theoretical justifications, 2) the significance of the problem we are trying to tackle, and 3) our careful effort in engineering skani.

Data & methodology: validity of approach, quality of data, quality of presentation

The approach is intuitive, as is the explanation of how it works. The evaluations are made across a broad range of different data sets (with different characteristics), which build confidence that the method produces robust results that compare well with those of existing methods.

The presentation is clear and the method is relatively straightforward to understand. I do think that the current manuscript would benefit from some extra detail (in the Extended Data or Online Methods)

describing important implementation details (i.e. what care had to be taken to make the method as fast as it is) and conveying what the current bottleneck is (i.e. is it computational, I/O, etc.).

R2.2. We have added the following subsection in the Methods section titled **skani implementation details** describing skani's implementation details:

Line 342:

skani is implemented in the rust, a systems-level programming language, for speed. skani implements four primary subcommands: sketch, dist, search, and triangle. The sketch command stores genomes in sketched representation (i.e. k-mer seeds and markers) for drop-in replacement for the other three commands. We use the same fast invertible k-mer hash function as minimap2 [41] for sketching. Additionally, skani uses AVX2 SIMD instructions to do vectorized seeding of 64-bit k-mers in 256-bit lanes when AVX2 instructions are detected, which we found to speed up sketching by approximately 30%. Chaining was implemented naively with no hardware accelerations.

dist, triangle, and search all calculate AF and ANI, but have different runtime behavior. dist and triangle keep all sketches in RAM, whereas the search command only keeps the marker k-mers in memory and allows for on-the-fly loading of the full sketch for each genome into RAM if the marker-based ANI threshold for the comparison is sufficiently high (80% by default), afterwards discarding the index from RAM. This makes search much more memory efficient and possibly more time-efficient than dist when querying a large database, but it is IO-bound and not as efficient as dist when performing all-to-all comparisons. The triangle command is similar to dist but is limited to computing distance matrices, but because skani is symmetric, it performs only $n(n - 1)/2$ comparisons instead of all n^2 . The inverted index for marker k-mer filtering is enabled by default in triangle, and otherwise in search and dist when the number of query genomes is ≥ 100 . See Extended Data Fig. 19 for a benchmark on skani's runtime scaling with thread usage.

Appropriate use of statistics and treatment of uncertainties

The method presented in this manuscript is deterministic. The relevant analyses seem appropriate.

Conclusions: robustness, validity, reliability

The conclusions drawn in the manuscript seem appropriate. Testing is performed across a variety of different datasets and across datasets conveying different characteristics. Thus, it appears these results are robust across a range of different settings and that the method generally performs well in terms of accuracy, runtime, and memory requirements.

Suggested improvements: experiments, data for possible revision

* All evaluations in the manuscript were carried out with 50 threads. While this is not an unreasonable number, and one might expect many users to run skani and other programs on large servers offering many concurrent threads of execution, I believe the manuscript would benefit from a scalability analysis on at least one of the large datasets being evaluated. How does runtime vary with the number of threads used? Does one expect that all of the methods would continue to scale well if more threads were available, or have some of them already saturated?

R2.3. We included a new scalability analysis in **Extended Data Fig. 19**. On the refseq-rc data set with about 4200 genomes, we timed the commands sketch, triangle, and search (described in the new section in **R2.2**). We see that the sketching and triangle commands stop scaling linearly at about 10-15 threads due to the IO time required to load and save sketches. The search command is much more IO bound, because it loads genomes into memory on-the-fly, and saturates earlier.

* From comparing the wall clock times to the CPU times, it seems that e.g. Mash is able to make better use of many (50) threads. Is there a reason that parallelization is easier there, or is this just an artifact of the current implementation.

R2.4. In Fig. 2, skani search does not scale well with the number of threads due to its high IO overhead, as discussed in **R2.3** and the new figure. Essentially all of the time for "skani search" in Fig. 2 is spent on IO. Mash loads much smaller indices into memory (only 1000 k-mers) so it can utilize the processing time better. However, mash triangle actually does not load all indices into memory and only loads on-the-fly, so like skani search, it is also mostly IO bound and does not scale as well as skani triangle (which loads all indices into memory).

Also, it would be useful to explore and describe what the relevant bottleneck step(s) currently are in skani. For example, is skani CPU limited during the chaining and chain extraction steps, or does it simply take longer to deserialize from the (larger) on-disk databases. If the latter is the case (i.e. if CPU utilization is substantially below all 50 threads), then perhaps it might make sense to trade some computation to reduce I/O overhead by, e.g., block-compressing the database on disk.

R2.5. In the new figure **Extended Data Fig. 19** and the new section in **R2.2**, we discuss some of the bottlenecks. Skani is indeed CPU limited during the chaining step, which is why triangle scales so well up to 10-15 threads, but deserializing time is non-trivial and mostly responsible for the drop-off in scaling.

We agree with the reviewer that it would make sense to trade some computation for IO overhead on the search step. A future idea would be some sort of memory mapped IO for very fast reading of the sketches. We have decided not to pursue this idea further for now, as the runtimes are still very reasonable.

* Likewise, the parameter c (the inverse of the fraction of selected seed k -mers) seems a critical one in skani. For example, the authors demonstrate (in Extended Data Figure 12) that in particularly challenging cases, where the default value of 125 struggles, skani with $c=30$ does very well. Is there some way to assess or determine when a particular value of c should be changed? On one hand, the larger c , the sparser the chaining and the faster it seems the algorithm will run. On the other hand, in more fragmented MAGs, a denser sampling can seemingly be more sensitive and lead to better ANI estimates. When should users consider changing the default? Is there any diagnostic that can be used to inform the user that the current value of c may be inappropriate, or at least a rule-of-thumb that can be used to select a reasonable c ?

R2.6. The referee is correct that c is an important parameter and we could do more to investigate it, and that fragmentation seems to be an important factor (as well as how distant the pairs of genomes are).

We give a new figure, **Extended Data Fig. 17** that explores the effect of N50 and fragmentation on ANI bias over various values of c . On four data sets, (Soil, Eukaryotic, Archaea, Pasolli) we plot regression curves of skani's ANI bias as a function of N50 over varying c . As expected, the smaller c , the better skani does on low N50 pairs of genomes.

In light of this new experiment, we added a new section in Methods called **Choosing the c parameter:**

Line 237:

The main parameter influencing runtime and accuracy is the c parameter. The default value of $c = 125$ works well on a variety of tasks such as searching databases and MAG comparison. We found that for very fragmented and distant genomes, lowering c may lead to more accurate ANI and AF estimates – see Extended Data Figs. 11, 12, 13, and 17. However, the runtime is inversely proportional to c . To guide users for choosing c , we suggest three different pre-set values of c (in addition to the default) in skani v0.1.2's help messages based on empirical heuristics: a “slow” pre-set with $c = 30$ for the most accurate AF estimates and pairs of genomes with $N50 \approx 3$ kb, a “medium” pre-set with $c = 70$ for genomes with $ANI \leq 95$ and $N50 \leq 10$ kb, and a “fast” pre-set $c = 200$ for similar genomes with $> 95\%$ ANI with $N50 > 10$ kb.

As stated, in the newest version of skani, we give three preset values of c called --slow, --medium, and --fast which are just aliases for $c = 30$, 70, and 200. The help messages in the software reiterate the empirical rules of thumb indicated in the text. The idea of having specific preset options is to guide users into a discrete set of choices, rather than a continuum, which may be hard to intuit.

We opted to not do automatic choosing of c parameters, because this will depend on the entire data set and is not trivial to implement. Furthermore, we think that a consistent value of c that gives more predictable performance, even if it is not optimal, is preferred—we don't want to overwhelm end-users with too many parameters.

References: appropriate credit to previous work?

Overall, the authors do a good job, in my opinion, of including appropriate references and citing them in the proper context.

One place where I think relevant references could (and should) be added is in the discussion of the chaining, chain scoring, and chain extraction sections. Specifically, the co-linear chaining algorithm introduced here is a standard approach. While Abouelhoda and Ohlebusch are appropriately cited, no citation is given for the "banded dynamic programming" heuristic used in skani. However, both the co-linear chaining formulation and the heuristic to speed up the co-linear chaining are very similar to what is used in minimap2 (Li 2018). There, chaining is accelerated by limiting the number of predecessors that will be considered for extending a chain given the intuition that, since predecessors are traversed in reverse order, from closest to farthest, once a good predecessor is found off of which to extend the current chain, it is unlikely that more distant predecessors will produce better scores. A similar relationship, though less direct, seems to hold between the approach used to extract minimally-overlapping chains in skani and the approach taken for a similar purpose in minimap2.

Finally, given the centrality of co-linear chaining to the skani approach, it seems worth to mention recent work on the co-linear chaining problem, like "Chaining with overlaps revisited" (Mäkinen and Sahlin 2020: <https://arxiv.org/abs/2001.06864v2>) and "Co-linear Chaining with Overlaps and Gap Costs" (Jain et al. RECOMB 2022). These describe increasingly practical $\tilde{O}(n)$ (where $\tilde{O}(\cdot)$ hides polylog(n) factors) co-linear chaining algorithms, and the latter paper also demonstrates important relationships between the edit distance and optimal co-linear chaining score that seem relevant, at least in theory, to the use herein of orthologous chains to estimate the ANI.

R2.7. We agree with the reviewer that chaining could be discussed in more detail.

In the "Chaining score function and algorithm" subsection, we have added the Makinen and Sahlin as well as the Jain et al. references as suggested.

Line 259:

The chaining problem can be solved optimally in sub-quadratic time for a variety of chaining costs [35,36], but we opt for a simple heuristic method that is fast and good enough for our purposes instead.

We agree with the author that our chaining procedure is quite similar to minimap2's. As such, we have added the following sentence and citation:

Line 269:

This banded procedure is a simpler version of minimap2.23's chaining procedure [37], which also employs a heuristic to stop chaining early via the `--max-chain-skip` parameter.

We did not describe minimap2's heuristic in more detail, because it may have changed since the initial publication.

The paper "Accelerating minimap2 for long-read sequencing applications on modern CPUs" by Kalikar et al describe the algorithm in minimap2.23 with the "max-chain-skip" parameter, with the following explanation: "According to the heuristic, if we do not find a better score over the last max_skip attempts, then the inner loop terminates. In minimap2, the default value for max_skip is 25.". This explanation was not related to optimal predecessors, only scores.

The documentation in the minimap2 manual describes this parameter as "A heuristics that stops chaining early [25]. Minimap2 uses dynamic programming for chaining. The time complexity is quadratic in the number of seeds. This option makes minimap2 exits the inner loop if it repeatedly sees seeds already on chains. Set INT to a large number to switch off this heuristics."

Nevertheless, the `--max-chain-skip` parameter in minimap2.23 very much related to our A parameter, as setting it to infinity gives a full dynamic programming (claimed by Kalikar et al. and the manual). We therefore refer to this parameter without mentioning any more detail, to avoid giving false methodological information dependent on minimap2 version implementation.

Clarity and context: lucidity of abstract/summary, appropriateness of abstract, introduction and conclusions

The abstract and conclusion are, in my opinion, clear and appropriate. The introduction describing and categorizing related work is brief, but appropriate in the context of the article type being submitted.

Other minor comments / typos

* Extended figure 2 should show the color bar legend.

R2.8. Extended Figure 2 (now Extended Fig. 1) now shows color bars.

* Extended Data figure 11 has a math mode escape issue in the caption.

R2.9. Fixed.

Reviewer response 3:

Remarks to the Author:

Shaw and Yu describe a tool called skani for accurate estimation of the ANI between two MAGs despite incompleteness and contamination. The selling points of the tool are its speed and its accuracy in the face of incompleteness/contamination. The accurate calculation of ANI is a topic of great interest. And results were convincing to me that the method is able to compute ANI more accurately than fast methods and faster than accurate methods. Thus, I find the work to be valuable overall. Below are some comments to improve the manuscript.

R3.1. We thank the reviewer for their evaluation of our work.

Major

1. There must be a limit at which any method of ANI calculation will start to fail due to incompleteness and/or contamination. Reading the paper, I didn't get any sense of what that level is for skani. I would like to see an experiment that tells me skani can still compute accurate distances as long as incompleteness is smaller than X or contamination is smaller than Y. Or something along those lines. Basically, I'd like to see the limits of the method in terms of input data quality. This analysis will be extremely helpful to readers and will save authors the headache in the future to explain to naive users

that such and such datasets should not have been used with this method, to begin with (e.g., two completely non-overlapping contigs of the same genome).

R3.2. We agree with the referee that we could make the exact regimes where skani operates optimally more clear.

Firstly, skani outputs an ANI estimate only if the AF is greater than 15% by default (can be changed as a parameter). FastANI uses the exact same idea, only with a 20% AF threshold. In practice, with default parameters, we get down to about 82% for the three data sets in Fig. 2. This can be changed with the `--min-aligned-frac` option in skani. We add the following text in the main body to make this more clear:

Line 136:

skani outputs an ANI estimate only if one of the genomes has predicted $AF \geq 15\%$ by default, which ends up giving reasonable ANIs down to the 82% range on the three data sets shown.

We also include an extended discussion on comparing low-ANI and incomplete genomes in our user manual for skani at <https://github.com/bluenote-1577/skani/wiki/skani-advanced-usage-guide#comparing-lower-ani-genomes> to explain to readers the limits of ANI comparison.

For contamination, we believe the new Extended Data Fig. 4 referenced in **R3.6** will also be useful for readers. Interestingly, contamination is not that big of an issue, because it seems like contaminants with $\leq 75\%$ ANI to the real genome do not get picked up by skani's chaining procedure, which is just not sensitive enough in this regime. So technically, as long as the contaminant genome is $< \sim 75\%$ ANI and the real genome is $> 15\%$ present, skani will output an ANI estimate.

2. It does not come through in the abstract and much of the intro that the focus is on very high ANI. A lot of us find low ANI values also interesting. This comes through later but should be clarified in the abstract and/or intro.

R3.3. We added the following text to the introduction to make it clear we are operating in the range of $> 82\%$ by default.

Line 45:

We developed skani, a fast, robust tool for calculating aligned fraction and ANIs in the > 82% range.

3. While most of the methodological steps were based on concepts I could easily accept, the idea that the under-estimation bias is solved using a trained model makes me somewhat worried, especially because that's the most interesting challenge in terms of accuracy. I ask for one clarification for line 291 below in minor comments. And I did note that the authors have only a small number of features, which is good. But, several other analyses would help me become more confident:

- * Show clearly the comparison between skani with and without this correction.

- * Show analyses where the bias correction model is trained on a dataset where *every* training MAG has low (e.g., < 80) ANI to *every* MAG in the testing set (or just train on archaea and apply on bacteria). Does the bias correction still work? If yes, I will feel very confident (I am not convinced by the vague "disjoint" assertion). If not, perhaps the method will not scale quite as well to entirely novel groups.

R3.4. For the first point "*Show clearly the comparison between skani with and without this correction.*" on showing skani with and without the correction, we have made the following changes:

1. We removed Extended Data Fig. 10 in the previous version and **changed Extended Data Fig. 13 in the new version by adding a new third row**, showing the relationship between skani ANI with no-learned regression.
2. We have **changed Extended Data Fig. 10 in the new version to display skani without regression as an additional violin plot**. The violin plot is unchanged for the other methods.

In the new figures, it is clear that the regression step always pulls down the ANI to be closer to the true value, but it is quite conservative.

For the second point, "*Show analyses where the bias correction model is trained on a dataset*", we performed a **new experiment shown in Extended Data Fig. 18**.

We added the following text to the Methods section:

Line 331:

To show that skani is not simply memorizing the organisms in the data set, we partitioned this training set into two sets, A and B, where each part is “disjoint” from one another in the sense that skani does not output an ANI estimate between any MAG in A and B due to low aligned fraction (< 15%). We show in Extended Data Fig. 18 that this debiasing procedure still corrects the ANI to be closer to a MUMmer-based ground truth.

We partitioned the training data set (Nayfach et al data set in Supp. Table 2) into two sets as follows.

1. We ran skani (without regression) and did all-to-all comparisons on the whole data set.
2. We connected each MAG to another MAG if skani outputs an ANI estimate (i.e. AF > 15%) in a graph structure
3. We sampled connected components in this graph until the total number of MAGs was more than half of the data set, call this set A.
4. We took this set A trained on it, and tested skani on all other MAGs in the data set (call this set B).

In this setup, A is completely “disjoint” from B as no MAG in A has > 15% AF to any MAG in B. We show in Extended Data Fig. 18 that under this set-up, the regression still works as expected, lowering the mean ANI deviation from ANIm towards 0.

4. Authors can do a better job of situating the work within the literature. They do a good job of covering methods focused on MAG ANI. But there is a distinct but related literature on ANI calculation from short reads. In particular, in that literature, the idea that incompleteness can mislead Jaccard was noted and studied in these previous studies (at least three, shown below). It would be necessary to discuss them. In particular, Afann uses machine learning to correct for low coverage in ways that may be related to the proposed methods. Skmer, in contrast, uses a formula, which is quite different from what is done here. I am not suggesting that an actual comparison to these methods is necessary (since they are designed for reads). But the conceptual connection should be made.

* Tang, Kujin, Jie Ren, and Fengzhu Sun. "Afann: Bias Adjustment for Alignment-Free Sequence Comparison Based on Sequencing Data Using Neural Network Regression." *Genome Biology* 20, no. 1 (December 4, 2019): 266. <https://doi.org/10.1186/s13059-019-1872-3>.

* Sarmashghi, Shahab, Kristine Bohmann, M. Thomas P. Gilbert, Vineet Bafna, and Siavash Mirarab. "Skmer: Assembly-Free and Alignment-Free Sample Identification Using Genome Skims." *Genome Biology* 20, no. 1 (December 13, 2019): 34. <https://doi.org/10.1186/s13059-019-1632-4>.

* Fan, Huan, Anthony R. Ives, Yann Surget-Groba, and Charles H. Cannon. "An Assembly and Alignment-Free Method of Phylogeny Reconstruction from next-Generation Sequencing Data." *BMC Genomics* 16, no. 1 (December 14, 2015): 522. <https://doi.org/10.1186/s12864-015-1647-5>.

R3.5. We agree with the reviewer that there is a nice conceptual connection between ANI/distance calculation between genomes using NGS reads and fragmented genomes, where the same "incompleteness" issue occurs but with random sampling being the cause instead of assembly fragmentation.

We have added the following text to the **Sequence identity estimation** subsection in Methods, mentioning the above methods (new text in red).

Line 161:

However, when dealing with MAGs, we don't have G and G' but instead fragmented, contaminated, and incomplete versions of G and G'. The models used in these sketching methods give biased estimates for ANI that are too small because missing k-mer matches may be due to mutations, incompleteness, or contamination instead of only due to mutations. **This issue is also present in the context of k-mer based alignment-free genome comparison using reads [27,28,29]. The same incompleteness issue is present due to a genome potentially not being fully covered due to the random sampling of reads. In our case, however, we have access to fragmented assemblies instead of reads.** Thus to accurately use k-mer statistics, we first find orthologous regions by approximate alignment, and then use k-mer statistics.

5. The simulation study was a bit too sparse in my opinion (real data analyses were great). Why only test 100% true ANI? Why not also throw mutations at random on a base genome with a known theta so that you can measure accuracy? Also, contamination effects were not studied. You can combine in some random contig from an unrelated genome and create a chimera.

R3.6. We added three new simulated experiments. First, we added two new experiments which are the same as the N50/Incompleteness heatmap (Extended Data Fig. 1) but with random substitutions so that the pairs of genomes are 95% and 90% similar (**new Extended Data Figs. 2,3**). We also added a chimeric MAG experiment (**new Extended Data Fig. 4**). We have also changed the experiment to be more realistic than the previous version: we now filter small contigs of length less than < 1kbp; previously we were generating small contigs that would realistically get filtered out by the MAG binning process.

The new Extended Data Fig. 2,3 shows that skani seems to be underestimating ANI for 95%, 90% ANI genomes, but this runs contrary to our experiments in Extended Data Fig. 13, 17, which shows the opposite effect for all of our data sets (real ANI deviation tends to slightly **overestimate** ANI). We believe that the discrepancy is most likely due to the random model not capturing nuances of real MAG binning/assembly. We explain in the caption of Extended Data Fig. 2 as follows:

Extended Data Fig.2 caption:

We performed the same simulation procedure as in Extended Data Fig. 1 and additionally induced random point substitutions so that the true pairwise ANI is 95%. Notably, skani underestimates ANI on the simulated data set whereas on real data sets slight overestimation of ANI is more common (Extended Data Fig. 13). This stems from skani's heuristic for dealing with fragmented assemblies, which removes non-homologous k-mers only on the ends of the chunks (see Estimating ANI from chains in Methods), not being as effective on i.i.d random fragments between two genomes. Importantly, fragmentation in real assemblies depends on the genomic loci characteristics (e.g. sequence repetitiveness), so is not i.i.d between two genomes.

While we think this is an interesting phenomenon and worthy of more investigation, we also believe it is out of the scope of this project.

The new Extended Data Fig. 4 compares the 5 algorithms on a chimeric contig of increasing chimericism. Interestingly, ANIu, due to its sensitivity and ability to calculate low-ANI pairs of genomes, has a lower ANI value than the other methods on these chimeric MAGs. skani and ANIm are less affected by this phenomenon. The lower sensitivity of skani and ANIm actually make it better at comparing contaminated MAGs.

Minor

* Figure SE: 5% error does not seem bad to readers who don't understand the applications of ANI. You should perhaps emphasize why 5% is such a higher error for ANI=100%. Describe applications where that error is catastrophic.

R3.7. We have added the following line to the introduction:

Line 59:

A difference of 4% is a large discrepancy for ANI, as it may cause two very similar genomes of up to 99% ANI to be classified as different species when subject to the standard 95% ANI species threshold [9].

* Lines 74-79. The authors say "phylogenetic observations" (which is somewhat oddly worded). But then, they use a non-phylogenetic method (average linkage clustering). Why not use something more phylogenetic like NJ or FASTME. It doesn't matter for these low distances but you may also want to do JC correction. This point is not crucial due to the low distances.

R3.8. We agree with the reviewer that emphasizing phylogeny was not the right choice of words, as we are not explicitly trying to examine evolutionary relationships between the MAGs. We are more interested in general types of downstream applications, such as visualization and MAG clustering. For example, in MAG dereplication, average-linkage clustering (as in dRep (Olm et al, 2017)) is often used because the output of interest is a clustering, and the exact evolutionary relationships are not important.

As such, we have rewritten the paragraph as follows to focus on heatmap/cluster comparison instead of phylogeny:

Line 77:

Because ANI underestimations due to MAG quality are systematically biased, such ANI estimates can strongly impact downstream applications. We show in Fig. 1c that the cluster heatmaps obtained by average linkage clustering for a species-level bin differ greatly between ANI methods. skani's heatmap qualitatively resembles ANIm's heatmap (Extended Data Fig. 5) more closely than the other methods, yet it is > 500 times faster than ANIm and > 50 times faster than FastANI for computing the distance matrix (Extended Data Fig. 8).

* Line 108: constant time with respect to what? The number of reference genomes? Something else? I am not even sure if this is true after reading the Methods. Make the claim more precise or remove it.

R3.9. We removed "near constant-time" and replaced it with "fast" on line 108.

Furthermore, we have removed the sentence on Line 114 with the "essentially constant time" in the original submission with the following sentence to make the scaling clear:

Line 118:

Given n genomes and s marker k -mers in a query, we can check all of its markers against the inverted index to calculate the max-containment. This is much faster than the $O(ns)$ naive linear-search time when the database is diverse and large.

* Line 127: The sentence reads too vague.

R3.10. The original sentence "A drawback of skani is that the index file sizes are larger than pure sketching methods (Supplementary Table 3), but disk space is usually less of an issue than RAM." has been replaced with

Line 132:

A drawback of skani is that the index file sizes are larger than pure sketching methods (Supplementary Table 3), **but the full indices can be stored on disk and only read into RAM by skani as needed.**

* Line 147: Not clear what you mean but "such spurious matches". Seems like a sentence is left out.

R3.11. We have rewritten the sentence in question to be more clear that "spurious" referred to repetitive k-mer matches not arising from sequence homology.

Line 157:

We proved in a previous work that for random, mutating strings, the expected number of k-mer matches arising spuriously from repetitiveness for a string of length n is $\sim n^2/4^k$ (Theorem 4 in Shaw and Yu [26]), so the usual assumption of no repetitive k-mers is not a bad one in practice for simpler, non-eukaryotic genomes and large enough k .

* Line 164: Would be good to give some idea here of how these k-mers are selected. Perhaps name the method (min hash)?

R3.12. We have included the reference to FracMinHash earlier on this line, which now reads

Line 182:

We extract $1/c$ fraction of k-mers for both genomes for some c ($c = 125$ by default) as seeds to be used for chaining using FracMinHash (section Obtaining sparse seeds for chaining).

* Line 159-170: Hard to connect steps 1-6 to the subsequent subheadings. Use keywords in steps or use step numbers in the heading to make it easy to connect.

R3.13. We have added section references to steps 1-6 in the "Algorithm outline" section, as requested. For example (additions in red):

Line 174-190:

We use a very sparse set of marker ℓ -mers to estimate max-containment index and obtain a putative ANI using the FracMinHash method (section Sketching by FracMinHash). We filter out pairs of genomes with putative ANI < 80% (section Max-containment putative ANI screening with marker \$\ell\$ -mers).

* Equation after line 189: I don't get why this is correct unless the part of the MAG covered by A is a subset of the parts covered by B (or vice versa). Or is this equation written only for the case where A and B are both complete genomes?

R3.14. The referee is correct that this equation only holds under the assumption that either A is contained in B or B is contained in A. We have changed the sentence to the following (addition in red):

Line 208:

Assuming one of the genomes is contained in the other completely, we then calculate an ANI estimate between two genomes G1, G2 as ...

* Truly sub-linear is too strong here. If I understand correctly, you compute the intersection with any genome that shares at least one marker kmer with the query. If not, please specify how you decide which references to compare against (a minimum number of markers needed)? If you do that, then, all it takes to increase the number of reference genomes is one "bad" marker (e.g., one that is ubiquitous). Since MinHash does not take into account things like the frequency and complexity of a kmer, there is no guard against this. I think, at a minimum, low complexity kmers should be disqualified as markers.

R3.15. We have replaced "truly sub-linear" with "fast" in **Line 218**. By sub-linear, we originally meant: if we had s markers and n genomes, it was faster than $O(ns)$ because you did not have to check s markers against all n k-mers using the inverted indexing procedure, but the referee is right in that "sub-linear" was not a precise and accurate way to state what is happening.

As an example: Suppose that a query had 998 unique markers (only shared with one other genome in the database), and 2 markers that were shared with all n genomes in the database. Then we hash all

1000 markers against the inverted index and count intersections. This time would take $O(998 + 2 * n)$ to get the containment index, which is much faster than the $O(1000 * n)$ operations for a naive search.

We agree that low-complexity k-mers should be disqualified as markers. In practice, we use minimap2's hash function (mentioned in the new section as a response to **R2.2**), which empirically seems to have good randomness properties (e.g. AAAAA... is not selected) and is battle-tested. We think there is room for using a weighted hash function, such as in "Weighted minimizer sampling improves long read mapping" by Jain et al, but we think this is out of the scope of our paper for now.

* Lines 249-255: Calling these orthologous is a stretch. If all goes well, this is just a closest-match procedure, which is not the same as orthology. I agree the distinction doesn't matter for you but why call it orthology and confuse readers/users?

R3.16. The reason for using the term "orthology" is to convey the fact that we want 1-to-1, minimally-overlapping matches between the two genomes, so that one region can not chain to multiple regions, and also that multiple regions can not chain to the same gene. Our minimally-overlapping greedy chain finding procedure gives us 1-to-1 mappings that do not overlap much. This is exactly the same output (1-to-1 minimally overlapping matches) as the result given by the reciprocal mapping approach used by OrthoANI (Lee et al, 2016), which emphasizes the "orthologous" nature of their approach.

We agree that this type of match lacks the same biological implications as "orthology", and that the chains are not base-level alignments, so lacks the sensitivity one desires from "orthology". However, the FastANI paper also uses the term "orthologous mapping" and uses k-mers as well.

Instead of removing the term completely, we have adding a warning, clarifying our usage of "orthology":

Line 283:

We will use the term "orthologous" loosely in the same sense as other ANI methods [9] –

we denote a set of mappings (i.e. chains) to be orthologous if they do not overlap too much along one of the genomes (i.e. no duplicated one-to-all mappings).

* Line 277: I am not sure what a percentile bootstrap is and how it differs from normal bootstrap. More broadly, I don't understand what object you are subsampling *with replacement*. What is the meaning of having sampled the same chunk twice? Perhaps this is not bootstrapping and just subsampling?

R3.17. By “percentile bootstrap”, we mean performing the standard bootstrap and then obtaining 90% confidence interval by taking the 5th and 95th percentile observations.

We took the terminology from Wikipedia:

[https://en.wikipedia.org/wiki/Bootstrapping_\(statistics\)#Deriving_confidence_intervals_from_the_bootstrap_distribution](https://en.wikipedia.org/wiki/Bootstrapping_(statistics)#Deriving_confidence_intervals_from_the_bootstrap_distribution). It is referred to as the “Percentile method” in “The jackknife, the bootstrap, and other resampling plans, In Society of Industrial and Applied Mathematics CBMS-NSF Monographs, 38.” by Efron (1982). To make it more clear, we rewrote the sentence as follows:

Line 312:

Because our final ANI estimate is just a weighted mean over the ANIs of the chunks, once we have these ANIs, we can quickly do bootstrapping by resampling the chunks and calculating the weighted mean over the resamples. We use 100 iterations and only proceed if there are > 10 chunks, outputting the 5th and 95th percentile ANIs to be our final 90% confidence interval. We found that the bootstrap gave reasonable ANI confidence intervals over skani’s inherent uncertainty (e.g. k-mer seeding variance), but we stress that it does not account for any systematic biases of skani’s ANI estimator relative to other methods.

We are sampling the ANIs (corresponding to the chunks) with replacement; there is no intrinsic meaning with sampling the same chunk (and thus, the same ANI) twice, but we are just interested in obtaining a quantification on the uncertainty of our measurement. We are not resampling chunks and then re-running skani, only resampling the output distribution given by the ANIs over the chunks.

* Line 291: Related to major comment 3: What does "disjoint" mean? Do they have high ANI to all the MAGs used for testing? Or just that it's a different dataset? In terms of "overfitting" (the subcontext of this paragraph), it doesn't matter whether the MAG is called the same or not if a MAG in the testing has low ANI to a MAG in training.

R3.18. We have removed the “These assembled MAGs are disjoint from the other sets of MAGs used in this study” line referenced in the original manuscript. The original meaning was that these are literally different assemblies than the other assemblies used in the study, but we agree that this point isn’t particularly useful. The new experiments and text outlined in **R3.4** addresses our original intention for the “disjoint” meaning.

* Latex error: ¿ 90% caption of Figure E11. Also, adding a horizontal line at zero helps the eye here.

R3.19. Fixed; same comment as **R1.6**.

Decision Letter, first revision:

Our ref: NMETH-BC51689A

5th Jun 2023

Dear Dr. Yu,

Thank you for submitting your revised manuscript "Fast and robust metagenomic sequence comparison through sparse chaining with skani" (NMETH-BC51689A). It has now been seen by the original referees and their comments are below. The reviewers find that the paper has improved in revision, and therefore we'll be happy in principle to publish it in Nature Methods, pending minor revisions to satisfy the referees' final requests and to comply with our editorial and formatting guidelines.

TRANSPARENT PEER REVIEW

Nature Methods offers a transparent peer review option for new original research manuscripts submitted from 17th February 2021. We encourage increased transparency in peer review by publishing the reviewer comments, author rebuttal letters and editorial decision letters if the authors agree. Such peer review material is made available as a supplementary peer review file. Please state in the cover

letter 'I wish to participate in transparent peer review' if you want to opt in, or 'I do not wish to participate in transparent peer review' if you don't. Failure to state your preference will result in delays in accepting your manuscript for publication.

ORCID

Sincerely,

Lin Tang, PhD
Senior Editor
Nature Methods

Reviewer #1 (Remarks to the Author):

I would like to thank the Authors for their thoughtful and thorough responses to my original review. I have started using skani and have found it to be robust, well-documented software.

Reviewer #2 (Remarks to the Author):

The authors have substantially revised the manuscript, added a considerable number of new experiments, and have improved the exposition of their contribution in the context of prior work. They have thoroughly addressed all of my concerns, and I congratulate them on the manuscript and tool. I believe the community will find skani very useful.

Reviewer #3 (Remarks to the Author):

The authors have done an excellent job of addressing my previous comments. I am satisfied with the revisions. In particular, I find extended figures 2 to 4 to be a great addition, and I find extended figure 18 very reassuring. I suggest some points to consider for *future work*. For this revision, the only remaining comment I have is that what is described as bootstrap *may* be better described as "block bootstrap". Please look up that terminology and see if that provides a better name.

- Figures E3 and E4 are interesting for the reasons you highlighted (under-estimation). I am not sure if the potential explanation is correct, but I think that would make a great topic for future investigation. If I have to guess, I would think that this underestimation does have some solution if you find the exact cause of it.

Author Rebuttal, first revision:

Point-by-point reviewer response.

Response legend:

Blue - Reviewer

Black - Authors.

Reviewer response 1:

Remarks to the Author:

I would like to thank the Authors for their thoughtful and thorough responses to my original review. I have started using skani and have found it to be robust, well-documented software.

R: We thank the reviewer for their questions and comments, which have greatly improved the clarity of our manuscript.

Reviewer response 2:

Remarks to the Author:

The authors have substantially revised the manuscript, added a considerable number of new experiments, and have improved the exposition of their contribution in the context of prior work. They have thoroughly addressed all of my concerns, and I congratulate them on the manuscript and tool. I believe the community will find skani very useful.

R: We thank the reviewer for their very helpful suggestions and feedback, especially on benchmarking skani.

Reviewer response 3:

Remarks to the Author:

The authors have done an excellent job of addressing my previous comments. I am satisfied with the revisions. In particular, I find extended figures 2 to 4 to be a great addition, and I find extended figure 18 very reassuring. I suggest some points to consider for *future work*. For this revision, the only remaining comment I have is that what is described as bootstrap *may* be better described as "block bootstrap". Please look up that terminology and see if that provides a better name.

R: We thank the reviewer for their helpful critiques that have substantially improved our experimentation.

We looked at the term “block bootstrap”, but unfortunately it seems like this term has been primarily used by the time-series forecasting statistical community. We have retained the original term to avoid this potential confusion.

- Figures E3 and E4 are interesting for the reasons you highlighted (under-estimation). I am not sure if the potential explanation is correct, but I think that would make a great topic for future investigation. If I have to guess, I would think that this underestimation does have some solution if you find the exact cause of it.

R: We agree with the reviewer that there may potentially be a better way of demonstrating our explanation. Although we think this is currently out-of-scope, skani is being continuously improved (four patches: v0.1.0 -> v0.1.4 in the past 4 months) and we may tackle this more seriously in the future.

Final Decision Letter:

22nd Aug 2023

Dear Professor Yu,

I am pleased to inform you that your Brief Communication, "Fast and robust metagenomic sequence comparison through sparse chaining with skani", has now been accepted for publication in Nature Methods. Your paper is tentatively scheduled for publication in our November print issue, and will be published online prior to that. The received and accepted dates will be 7th Feb 2023 and 22nd Aug 2023. This note is intended to let you know what to expect from us over the next month or so, and to let you know where to address any further questions.

Over the next few weeks, your paper will be copyedited to ensure that it conforms to Nature Methods style. Once your paper is typeset, you will receive an email with a link to choose the appropriate publishing options for your paper and our Author Services team will be in touch regarding any additional information that may be required.

Your paper will now be copyedited to ensure that it conforms to Nature Methods style. Once proofs are generated, they will be sent to you electronically and you will be asked to send a corrected version within 24 hours. It is extremely important that you let us know now whether you will be difficult to contact over the next month. If this is the case, we ask that you send us the contact information (email, phone and fax) of someone who will be able to check the proofs and deal with any last-minute problems.

Once your manuscript is typeset and you have completed the appropriate grant of rights, you will receive a link to your electronic proof via email with a request to make any corrections within 48 hours. If, when you receive your proof, you cannot meet this deadline, please inform us at rjsproduction@springernature.com immediately.

Once your paper has been scheduled for online publication, the Nature press office will be in touch to confirm the details.

Content is published online weekly on Mondays and Thursdays, and the embargo is set at 16:00 London time (GMT)/11:00 am US Eastern time (EST) on the day of publication. If you need to know the exact publication date or when the news embargo will be lifted, please contact our press office after you have submitted your proof corrections. Now is the time to inform your Public Relations or Press Office about your paper, as they might be interested in promoting its publication. This will allow them time to prepare an accurate and satisfactory press release. Include your manuscript tracking number NMETH-BC51689B and the name of the journal, which they will need when they contact our office.

About one week before your paper is published online, we shall be distributing a press release to news organizations worldwide, which may include details of your work. We are happy for your institution or funding agency to prepare its own press release, but it must mention the embargo date and Nature Methods. Our Press Office will contact you closer to the time of publication, but if you or your Press Office have any inquiries in the meantime, please contact press@nature.com.

Please note that *Nature Methods* is a Transformative Journal (TJ). Authors may publish their research with us through the traditional subscription access route or make their paper immediately open access through payment of an article-processing charge (APC). Authors will not be required to make a final decision about access to their article until it has been accepted. [Find out more about Transformative Journals](https://www.springernature.com/gp/open-research/transformative-journals)

To assist our authors in disseminating their research to the broader community, our SharedIt initiative provides you with a unique shareable link that will allow anyone (with or without a subscription) to read the published article. Recipients of the link with a subscription will also be able to download and print the PDF. As soon as your article is published, you will receive an automated email with your shareable link.

Please note that you and your coauthors may order reprints and single copies of the issue containing your article through Springer Nature Limited's reprint website, which is located at <http://www.nature.com/reprints/author-reprints.html>. If there are any questions about reprints please send an email to author-reprints@nature.com and someone will assist you.

Please feel free to contact me if you have questions about any of these points. Thank you very much for publishing your paper at Nature Methods!

Best regards,

Lin Tang, PhD
Senior Editor
Nature Methods

** Visit the Springer Nature Editorial and Publishing website at <http://www.springernature.com/editorial-and-publishing->

jobs?utm_source=ejp_NMeth_email&utm_medium=ejp_NMeth_email&utm_campaign=ejp_Nmeth">www.springernature.com/editorial-and-publishing-jobs for more information about our career opportunities. If you have any questions please click here.**